# Attention Interpolation for Text-to-Image Diffusion

Qiyuan He[1]     Jinghao Wang[2]     Ziwei Liu[2]     Angela Yao[1, ✉]

[1]National University of Singapore     [2]S-Lab, Nanyang Technological University

qhe@u.nus.edu.sg     ayao@comp.nus.edu.sg

{jinghao003, ziwei.liu}@ntu.edu.sg

## Abstract

Conditional diffusion models can create unseen images in various settings, aiding image interpolation. Interpolation in latent spaces is well-studied, but interpolation with specific conditions like text or image is less understood. Common approaches interpolate linearly in the conditioning space but tend to result in inconsistent images with poor fidelity. This work introduces a novel training-free technique named **Attention Interpolation via Diffusion (AID)**. AID has two key contributions: **1)** a fused inner/outer interpolated attention layer to boost image consistency and fidelity; and **2)** selection of interpolation coefficients via a beta distribution to increase smoothness. Additionally, we present an AID variant called **Prompt-guided Attention Interpolation via Diffusion (PAID)**, which **3)** treats interpolation as a condition-dependent generative process. Experiments demonstrate that our method achieves greater consistency, smoothness, and efficiency in condition-based interpolation, aligning closely with human preferences. Furthermore, PAID offers substantial benefits for compositional generation, controlled image editing, image morphing and image-controlled generation, all while remaining training-free. Our code and demo are available at https://qy-h00.github.io/attention-interpolation-diffusion/.

## 1 Introduction

Interpolation is a common operation applied to generative image models. It generates smoothly transitioning sequences of images from one seed to another within the latent space and facilitates applications in image attribute modification [40], data augmentation [38], and videos [46]. Interpolation has been investigated extensively [18, 43, 42] in VAEs [20], GANs [8], and diffusion models [13]. Text-to-image diffusion models [35, 37] are a new class of *conditional* generative models that generate high-quality images conditioned on textual descriptions. How to interpolate between distinct text conditions such as "a truck" and "a cat" (see Fig. 1 (d)) is relatively under-explored. This issue is, however, crucial for various downstream tasks, such as conditional generation with multiple conditions [6, 25, 51] or cross-modality conditions [54, 56], as well as for image editing [11, 48], where precise control over impact of different conditions is essential to achieve desired results.

This paper formulates the task of *conditional interpolation* and identifies three ideal properties for interpolating text-to-image diffusion models: thematic consistency, smooth visual transitions between adjacent images, and high-quality interpolated images. For instance, interpolating from *"a truck"* to *"a cat"* should avoid irrelevant transitions (*e.g.*, via *"a bowl"*). The sequence should change between the two conditions gradually and feature high-quality and high-fidelity images (*vs. e.g.* simple overlays of the truck and cat). These properties directly motivate our quantitative evaluation metrics for conditional interpolation: consistency, smoothness, and fidelity.

A direct approach to traverse the conditioning space is interpolating in the text embedding itself [53, 55, 16]. Such an approach often has sub-optimal results (see the first row of Fig. 2). A closer analysis reveals that interpolating the text embedding is mathematically equivalent to interpolating the keys and values of the cross-attention module between the text and image space. Our analysis further

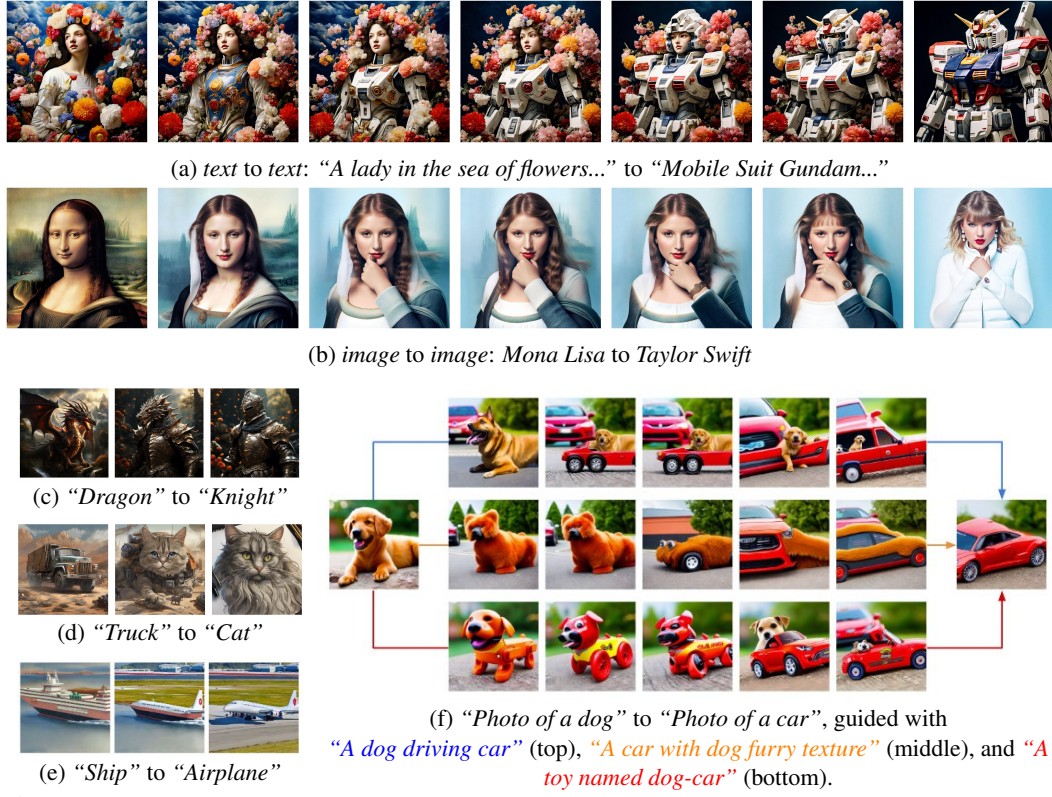

(a) *text* to *text*: *"A lady in the sea of flowers..."* to *"Mobile Suit Gundam..."*

(b) *image* to *image*: *Mona Lisa* to *Taylor Swift*

(c) *"Dragon"* to *"Knight"*

(d) *"Truck"* to *"Cat"*

(e) *"Ship"* to *"Airplane"*

(f) *"Photo of a dog"* to *"Photo of a car"*, guided with *"A dog driving car"* (top), *"A car with dog furry texture"* (middle), and *"A toy named dog-car"* (bottom).

Figure 1: **Our approach enables text-to-image diffusion models to generate nuanced spatial and conceptual interpolations between different conditions including text (a, c-e) and image (b),** with seamless transitions in layout, conceptual blending, and user-specified prompts to guide the interpolation paths (f).

reveals that the keys and values in self-attention impose a stronger influence than cross-attention, which may explain why text embedding interpolation fails to produce consistent results.

Based on our analysis, we introduce a novel framework: Attention Interpolation of Diffusion (AID) models for conditional interpolation. AID enhances interpolation quality with (1) a fused interpolated attention mechanism on both cross-attention and self-attention layers to improve consistency and fidelity and (2) a Beta-distribution-based sample selection along the interpolation path for interpolation smoothness. Additionally, we introduce (3) Prompt-guided Attention Interpolation of Diffusion (PAID) models to further guide the interpolation via a text description of the path itself.

Experiments on various state-of-the-art diffusion models [30, 35, 2] highlight our approach's effectiveness (see samples in Fig. 1 and more in Appx. H) without any additional training.

Human evaluators predominantly prefer our method over standard text embedding interpolation. We further show that our method can benefit various downstream tasks, such as composition generation, and boost the control ability of image editing. Our method is also compatible with image condition (see Fig. 1 (b)), which can be further used for more application such as image morphing and image-controlled generation. This underscores the practical impact of the problem of conditional interpolation and our proposed solution. Our main contributions are:

- Problem formulation for conditional interpolation within the text-to-image diffusion model context and proposing evaluation metrics for consistency, smoothness, and fidelity;

- A novel and effective training-free method AID for text-to-image interpolation. AID can be augmented with prompt-guided interpolation (PAID) to control specific paths between two conditions;

- Extensive experiments highlight AID's improvements for text-based image interpolation. AID substantially improves interpolation sequences, with significant enhancements in fidelity, consistency, and smoothness without any training. Human studies show a strong preference for the AID;

- We show that AID offers much better control ability for diffusion-based image editing, and it can be used for compositional generation with state-of-the-art performance. It is also compatible with image condition, enabling more applications such as image morphing or controlling the scale of additional image prompt.

## 2 Related Work

**Diffusion Models and Attention Manipulation**. The emergence of diffusion models has significantly transformed the text-to-image synthesis domain, with higher quality and better alignment with textual descriptions [35, 37, 33]. Attention manipulation techniques have been instrumental in unlocking the potential of diffusion models, particularly in applications such as in-painting and compositional object generation. These applications benefit from refined control over the attention maps, aligning the modifier and the target object more closely to enhance image coherence [11, 1, 3, 51, 34]. Furthermore, cross-frame attention mechanisms have shown promise in augmenting visual consistency within video generation frameworks utilizing diffusion models [17, 31]. These works suggest that the visual closeness of two generated images may be reflected in the similarity of their attention maps and motivates us to study interpolation from an attention perspective.

**Interpolation in Image Generative Models**. Interpolation within the latent spaces of models such as GANs [8] and VAEs [20] has been studied extensively [43, 18, 46]. More recently, explorations of diffusion model latent spaces allow realistic interpolations between real-world images [38, 21]. Works to date, however, are limited to a single condition, and there is a lack of research focused on interpolation under varying conditions. Wang & Golland explored linear interpolation within text embedding to interpolate real-world images; however, this approach yields image sequence with diminished fidelity and smoothness. This gap underscores the need for further exploration of conditional interpolation in generative models.

## 3 Analysis of Conditional Interpolation

### 3.1 Text-to-Image Diffusion Models

Text-to-image diffusion models such as Stable Diffusion [35, 30] generate images from specified text. Consider the generation of an image for some specified text as an inference process denoted by $f(z^T, c)$. The function $f$ is an abstraction representing the denoising diffusion process, $c$ is a representation of the conditioning text and $z^T$ is a randomly sampled latent seed. Usually, $c$ is represented as a CLIP text embedding [32], while the $z$'s over the denoising time steps of the generation are sampled from a Gaussian distribution. More specifically, if the inference is carried out over $T$ denoising time steps, the latent $z^{t-1}$ can be sampled conditionally, based on $z^t$:

$$p_\theta(z^{t-1}|z^t) = \mathcal{N}(z^{t-1}; \mu_\theta(z^t, c, t), \Sigma_\theta(z^t, c, t)), \quad \text{where } z^t \sim N(0, 1), \quad (1)$$

where $t$ represents the denoising time-step index, $\mu_\theta(z^t, c, t)$ is estimated by a UNet [36], and $\Sigma_\theta(z^t, c, t))$ is determined by a noise scheduler [13, 42]. After iterative sampling from $z^T$ to $z^0$, the image is generated by decoder $D$, as $D(z^0)$.

Attention is used in text-to-image diffusion models [35, 29, 37] in various forms. Cross-attention is commonly used as the link from the text condition to the image generation. Specifically, given a latent variable $z \in \mathbb{R}^{d_z}$, text condition $c \in \mathbb{R}^{d_c}$ and the attention layer with matrices $W_Q \in \mathbb{R}^{d_z \times d_q}$, $W_K \in \mathbb{R}^{d_c \times d_k}$ and $W_V \in \mathbb{R}^{d_c \times d_v}$, the cross-attention is computed as

$$A(z, c) = \text{Attn}(Q, K, V) = \text{softmax}(\frac{QK^{\mathsf{T}}}{\sqrt{d_k}})V, \quad \text{where } Q = W_Q^{\mathsf{T}}z, \ K = W_K^{\mathsf{T}}c, \ V = W_V^{\mathsf{T}}c. \quad (2)$$

Self-attention is also commonly used in state-of-the-art text-to-diffusion models [30, 35, 2]. Self-attention is a special case of cross-attention and can also be computed with Eq. 2 as $A(z, z)$. In this case, the key and values are defined as $K = W_K^{\mathsf{T}}z, \ V = W_V^{\mathsf{T}}z$ respectively. For brevity, we abuse the notation to directly represent multi-head attention [47] and denote the attention layer as $\text{Attn}(Q, K, V)$ in both cross-attention and self-attention scenarios.

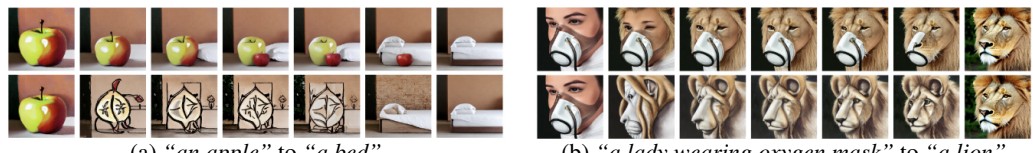

| (a) *"an apple"* to *"a bed"* | (b) *"a lady wearing oxygen mask"* to *"a lion"* |

Figure 2: **Results comparison between AID (the 1st row) and text embedding interpolation (the 2nd row).** AID increases smoothness, consistency, and fidelity significantly.

## 3.2 Text Embedding Interpolation

In this paper, we denote linear and spherical interpolation [49, 21] as $r_l(w; A, B)$ and $r_s(w; A, B)$ respectively, where $w \in [0, 1]$ is the interpolation coefficient, and $(A, B)$ are the interpolation anchors or end-points. Conditional interpolation differs from standard text-to-image generation in that there are two text conditions $c_1$ and $c_m$. Each condition has its own respective latent seeds $z_1$ and $z_m$[1]. The objective of conditional interpolation is to generate a sequence of images $\{I_{1:m}\} = \{I_1, I_2, ..., I_m\}$. In this sequence, the source images are generated by the standard text-to-image generation, i.e, $I_1 = f(z_1, c_1)$, $I_m = f(z_m, c_m)$ as described in Sec. 3.1.

Existing literature has shown that similarity in input space, including the latent seed and the embedding of the condition reflects the similarity in the output pixel space [19, 49]. Directly interpolating the text embedding $c$ is therefore a straightforward approach that can be used to generate an interpolated image sequence [49, 16]. In text embedding interpolation, the text conditions $\{c_1, c_m\}$ and their latent seeds $\{z_1, z_m\}$ are used as endpoints for interpolation and images are generated accordingly:

$$I_i = f(z_i, c_i) \text{ where } z_i = r_s(w_i; z_1, z_m), \ c_i = r_l(w_i; c_1, c_m), \text{ and } w_i = \frac{i-1}{m-1} \text{ for } i = \{1 \ldots m\}. \quad (3)$$

Note that spherical interpolation is applied to estimate the latent seed $z_i$ to ensure that Gaussian noise properties are preserved [38]. In contrast, linear interpolation is applied to estimate the text condition [53, 49, 16, 24] $c_i$. For both $z_i$ and $c_i$, the interpolation coefficient $w_i$ sampled in uniform increments from 1 to $m$.

Given that the text condition $c$ directly propagates into the key and value $K$ and $V$ (Eq. 2), interpolating between the text conditions $c_1$ and $c_m$ is equivalent to interpolating the associated keys and values in cross-attention. This is stated formally in the following proposition:

**Proposition 1.** *Given query $Q$ from a latent variable $z$, keys and values $\{K_1, V_1\}$ and $\{K_m, V_m\}$ from text conditions $\{c_1, c_m\}$ and linearly interpolated text conditions $c_i$, the resulting cross-attention module $A(z, c_i)$ is given by linearly interpolated keys and values $\bar{K}_i$ and $\bar{V}_i$:*

$$A(z, c_i) = Attn(Q, \bar{K}_i, \bar{V}_i), \text{ where } \bar{K}_i = r_l(w_i; K_1, K_m) \text{ and } \bar{V}_i = r_l(w_i; V_1, V_m), \quad (4)$$

*where $w_i$ is defined similarly as Eq. 3.*

The proof for Proposition 1 is given in the Appx. A. This proposition gives insight into how text embedding interpolation can be viewed as manipulating the keys and values. Specifically, it is equivalent to interpolating the keys and values to generate the resulting interpolated image. It is worth noting that an analogous interpretation does not carry through for the query $Q$ even though it also depends on some interpolated latent seed $z^t$. This is because $z_i^t$ is estimated as an interpolation between the latent seeds $z_1^t$ and $z_m^t$, while the latent $z_i^t$ itself is progressively altered through the denoising process (see Eq. 1).

## 3.3 Measuring the Quality of Conditional Interpolation

Text embedding interpolations work well when the conditions are semantically related, e.g., *"a dog"* and *"a cat"*, but may lead to failures in less related cases. To better analyze the characteristics of the interpolated image sequences, we define three measures based on ideal interpolation qualities: consistency, smoothness, and image fidelity.

---

[1]We use subscripts on $z$ to denote latent seed indices without expressing the $T$ denoising timesteps explicitly, i.e. $z_1^T = z_1$ and $z_m^T = z_m$.

**Perceptual Consistency.** Ideally, the interpolated image sequence should transition from one source or endpoint to the other in a perceptually direct and therefore consistent path. Similar to [15], we use the average LPIPS metric [57] across all adjacent image pairs in the sequence to evaluate consistency. If $P$ denotes the LPIPS model, the consistency $C$ of a sequence $I_{1:m}$ is defined as:

$$C(I_{1:m}; P) = \frac{1}{m-1} \sum_{i=1}^{m-1} P(I_i, I_{i+1}). \tag{5}$$

For example, a consistent interpolation from *"an apple"* to *"a bed"* may pass through *"an apple and a bed"* but should not have intermediate stages like *"a messy sketch"* (see Fig. 2 (a)).

**Perceptual Smoothness**. A well-interpolated sequence should exhibit a gradual and smooth transition. We propose to apply Gini coefficients on the perceptual distance between each neighbouring pair of interpolated images to indicate smoothness. Gini coefficients [5] are a conventional indicator of data imbalance [45, 9, 10, 50] where higher coefficients indicate more imbalance. And the imbalance of perceptual distance of each neighbouring pair indicates low smoothness. Let $G(X)$ denote the Gini coefficient of a set $X = \{x_1, x_2, ..., x_n\}$. The smoothness $S$ of a sequence $I_{1:m}$ with $P$ denoting the LPIPS model is defined as:

$$S(I_{1:m}; P) = 1 - G(\bigcup_{i=1}^{m-1} P(I_i, I_{i+1})), \quad G(X) = \frac{\sum_{i=1}^{n} \sum_{j=1}^{n} |x_i - x_j|}{2n \sum_{i=1}^{n} x_i}. \tag{6}$$

Fig. 2(b) shows how a smooth interpolation sequence exhibits a gradual transition on the visual content (from *"a lady wearing oxygen mask"* to *"lion"* in the top row) instead of one source image or end-point dominating the sequence (the *"lion"* in the bottom row).

**Fidelity**. Finally, any interpolated images should be of the same (high) quality conventionally generated images. Following [38, 49], we evaluate the fidelity of interpolated images with the Fréchet Inception Distance (FID) [12]. Given $n$ interpolated sequences $\{I_{1:m}^{(1)}, I_{1:m}^{(2)}, ..., I_{1:m}^{(n)}\}$, the fidelity $F$ of the sequences is defined as the FID based on a visual inception model [2] The FID between the source images and the interpolated images is defined as:

$$F(I_{1:m}^{(1)}, I_{1:m}^{(2)}, ..., I_{1:m}^{(n)}) = FID_{M_V} \left( \bigcup_{j=1}^{n} \{I_1^{(j)}, I_m^{(j)}\}, \bigcup_{j=1}^{n} \{I_i^{(j)} | i \neq 1, i \neq m\} \right) \tag{7}$$

For example, the interpolated sequence should have minimal artifacts (see Fig. 2(a)), where the top row clearly shows the appearance of the apple, whereas the bottom row does not.

### 3.4 Diagnosing Text Embedding Interpolation

Experimentally (see Sec. 5.1), we observe that text embedding interpolation sequences exhibit poor consistency and smoothness. The interpolated images are also commonly low in fidelity, with indirect and non-smooth transitions. *Where do the failures of text embedding interpolation come from?* We analyze the outputs from the perspective of spatial layouts and the selection of interpolation coefficients.

**Spatial Layouts and Attention.** Consistency is directly affected by the difference between the spatial layout of the source and interpolated images. One observation is that the spatial layout of interpolated images from text embedding interpolations is quite different from the source endpoints (see Fig. 2 (a) bottom row). Proposition 1 links the text embedding interpolation to the cross-attention mechanism exclusively. However, the literature suggests that the spatial layout of the overall image is strongly linked to the self-attention mechanism [17, 31]. As such, we hypothesize that cross-attention does not pose enough spatial layout constraints stand-alone. Instead, there is a need for a stronger link of the interpolation to self-attention, to allow more consistent spatial transitions.

As a simple test, we swap the keys and values from two text-to-image generations. Consider two images $I$ and $I'$ generated from two text prompts $p$ and $p'$. We replace the keys and values from

---

[2]Typically, Inception v3 [44] $M_V$ is used.

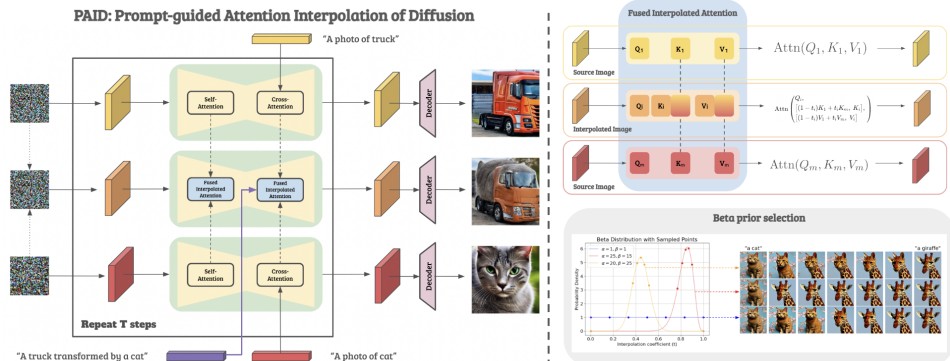

Figure 3: **An overview of PAID: Prompt-guided Attention Interpolation of Diffusion.** The main components include: (1) Replacing both cross-attention and self-attention when generating interpolated image by fused interpolated attention; (2) Selecting interpolation coefficients with Beta prior; (3) Inject prompt guidance in the fused interpolated cross-attention.

either the cross-attention or self-attention layers in the generative process of $I$ with that of $I'$ to generate $I_{\text{cross}}$ and $I_{\text{self}}$ respectively. We then evaluate the mean squared error (MSE) of the low-frequency components between $I'$ and $I_{\text{cross}}$ or $I_{\text{self}}$. The results show that $I_{\text{self}}$ closely resembles $I'$, while $I_{\text{cross}}$ does not. More details and results are shown in Appx. B.

**Selection of Interpolation Coefficients.** Interpolation methods [52, 12, 38] commonly select uniformly spaced coefficients $w_i$ on the interpolation path. Yet an observation from Fig. 2 (b) shows that uniformly spaced points in the text-embedding space do not lead to uniformly spaced images with smooth transitions. Small visual transitions may occur over a large range of interpolation coefficients and vice versa, which we can show quantitatively by comparing the perceptual distances between adjacent pairs of uniformly spaced coefficients. This suggests that we should adopt non-uniform selection to ensure smoothness. More details and results are shown in the Appx. B.

## 4   AID: Attention Interpolation of Text-to-Image Diffusion

The diagnosis in Sec. 3.4 directly leads us to make the following proposals for improving conditional interpolation. First, we are motivated to extend attention interpolation beyond cross-attention to self-attention as well (Sec. 4.1) and propose fused attention. Secondly, our diagnosis of the smoothness motivates us to adopt a non-uniform selection of interpolation coefficients to encourage more even transitions (Sec. 4.2). Combining these two techniques, we propose a **AID**: Attention Interpolation of text-to-image Diffusion.

Finally, in an effort to give more precise control over the interpolation path, we introduce the use of prompt guidance for interpolation (Sec. 4.3). This further enhances AID as Prompt-guidance AID (PAID). The full pipeline is shown in Fig. 3.

### 4.1   Fused Interpolated Attention Mechanism

The analysis in Sec. 3.4 highlights that both cross-attention and self-attention likely play a role in interpolating spatially consistent images. Proposition 1 can be generalized to self-attention where the keys and values are derived from the latent $z$ instead of $c$ for enhancing spatial constraint. As such, we define a general form of **inner-interpolated** attention on the keys and values as follows:

$$\text{Intp-Attn}_I\Big(Q_i, K_{1:m}, V_{1:m};\ w_i\Big) = \text{Attn}\Big(Q_i,\ (1-w_i)K_1 + w_i K_m,\ (1-w_i)V_1 + w_i V_m\Big),\ \text{(8)}$$

where $Q_i$ is derived from $z_i$. Note that Eq. 8 is equivalent to Eq. 4 if $\{K_1, K_m\}$ and $\{V_1, V_m\}$ are derived from $\{c_1, c_m\}$, i.e. as cross-attention; if they are derived from $\{z_1, z_m\}$, then it represents self-attention.

Instead of applying interpolation to the key and value, we can also interpolate the attention itself. We define this as **outer-interpolated attention**:

$$\text{Intp-Attn}_O\Big(Q_i, K_{1:m}, V_{1:m}; w_i\Big) = (1-w_i) \cdot \text{Attn}\Big(Q_i, K_1, V_1\Big) + w_i \cdot \text{Attn}\Big(Q_i, K_m, V_m\Big).\ \text{(9)}$$

Similarly, Eq. 9 can represent both cross- and self-attention, depending on if $\{K_1, K_m\}$ and $\{V_1, V_m\}$ are derived from $\{c_1, c_m\}$ or $\{z_1, z_m\}$ respectively. More details on the differences between inner and outer interpolation are given in the Appx. C. We denote the two versions as AID-I and AID-O for inner and outer interpolation respectively.

While applying interpolation as defined in Eqs. 8 and 9 for self-attention does lead to high spatial consistency, it also results in poor fidelity images. This is likely because directly replacing the self-attention mechanism with some interpolated version is too aggressive. Therefore, for self-attention, we maintain the source keys and values $K_i$ and $V_i$ from the interpolated $z_i$ and concatenate them with the interpolated keys and values, as shown in Fig. 3. Denoting concatenation as $[\cdot, \cdot]$, we define fused attention interpolation, leading to a **fused inner-interpolated attention**:

$$\text{Intp-Attn}_I^F\Big(Q_i, K_{1:m}, V_{1:m}; w_i\Big) = \text{Attn}\Big(Q_i, \; \big[(1-w_i)K_1 + w_i K_m, \; K_i\big], \; \big[(1-w_i)V_1 + w_i V_m, \; V_i\big]\Big). \tag{10}$$

For self-attention, as $K_i$ is derived from $z_i$, $K_i \neq (1-w_i)K_1 + w_i K_m$; the same holds for $V_i$. For cross-attention, however, $K_i = (1-w_i)K_1 + w_i K_m$, so fusing the two does not provide additional benefits. We note that there are opportunities for fusion with keys and values derived from other sources. We follow such a strategy in Sec. 4.3 to inject additional text-based guidance.

Analogous to Eq. 9, we define a **fused outer-interpolated attention**:

$$\text{Intp-Attn}_O^F\Big(Q_i, K_{1:m}, V_{1:m}; w_i\Big) = (1-w_i)\cdot\text{Attn}\Big(Q_i, [K_1, K_i], [V_1, V_i]\Big)$$
$$+ w_i\cdot\text{Attn}(Q_i, [K_m, K_i], [V_m, V_i]\Big). \tag{11}$$

## 4.2 Non-Uniform Interpolation Coefficients

The analysis in Sec. 3.4 shows that interpolation coefficients should not be selected uniformly adopted in previous methods [11, 16] on the interpolation path. For more flexibility, we apply a Beta distribution $p_B(t, \alpha, \beta)$. Beta distributions are conveniently defined within the range of $[0, 1]$. When $\alpha = 1$ and $\beta = 1$, $p_B$ degenerates to a uniform distribution, which reverts to the original setting. When $\alpha > 1$ and $\beta > 1$, the distribution is concave (bell-shaped), with higher probabilities away from the end-points of 0 and 1, i.e. away from the source images. Finally, the selected points are adjustable based on alpha and beta values, to give higher preference towards one or the other source image (see Fig. 3).

Given the Beta prior represented as cumulative distribution function $F_B(w, \alpha, \beta)$, we define a Beta-interpolation $r_B(w; 0, 1)$ as $r(F^{-1}(w, \alpha, \beta))$, where $w \sim U(0, 1)$. Therefore, the distributed point with Beta prior becomes:

$$\{r(0), r(F_B^{-1}(\frac{1}{m-1}, \alpha, \beta)), ..., r(F_B^{-1}(\frac{m-2}{m-1}, \alpha, \beta)), r(1)\}. \tag{12}$$

In practice, we employ a dynamic selection process to adjust the $\alpha$ and $\beta$ parameters of the Beta prior, and form the smoothest sequence from the explored coefficients. Further details are provided in Appendix D.

## 4.3 Prompt Guided Conditional Interpolation (PAID)

Given two source inputs, the hypothesis space of interpolation paths is actually large and diverse. Yet most interpolation methods [52, 38] estimate one deterministic path. Can we control or specify the interpolation path? One possibility is to provide a (third) conditioning text, which we refer to as a *guidance prompt*. To connect the interpolated sequence with the text in the guidance prompt $g$, we fuse the associated key $K_g = W_K^\mathsf{T} g$ and value $V_g = W_V^\mathsf{T} g$ instead of the original $K_i$ and $V_i$ in the fused inner-interpolated attention in Eq. 10 for cross-attention:

$$\text{Guide-Attn}_I^F\Big(Q_i, K_{1:m}, V_{1:m}; w_i, K_g, V_g\Big) =$$
$$\text{Attn}\Big(Q_i, \; \big[(1-w_i)K_1 + w_i K_m, \; K_g\big], \; \big[(1-w_i)V_1 + w_i V_m, \; V_g\big]\Big). \tag{13}$$

In practice, the guidance prompt is provided by users to choose the interpolation path conditioned on the text description as Fig. 1 (f) shows. We demonstrate that the prompt-guided attention interpolation dramatically boosts the ability of compositional generation in Sec. 5.2.

| Dataset | Method | Smoothness (↑) | Consistency (↓) | Fidelity (↓) |
|---|---|---|---|---|
| CIFAR-10 | TEI | 0.7531 | 0.3645 | 118.05 |
| | DI | 0.7564 | 0.4295 | 87.13 |
| | AID-O | 0.7831 | 0.2905* | 51.43* |
| | AID-I | 0.7861* | 0.3271 | 101.13 |
| LAION-Aesthetics | TEI | 0.7424 | 0.3867 | 142.38 |
| | DI | 0.7511 | 0.4365 | 101.31 |
| | AID-O | 0.7643 | 0.2944* | 82.01* |
| | AID-I | 0.8152* | 0.3787 | 129.41 |

(a)

| Interpolated attention | Self-fusion | Beta prior | Smoothness (↑) | Consistency (↓) | Fidelity (↓) |
|---|---|---|---|---|---|
| ✗ | ✗ | ✗ | 0.7531 | 0.3645 | 118.05 |
| ✗ | ✗ | ✓ | 0.7995 | 0.3803 | 117.30 |
| ✓ | ✗ | ✗ | 0.7846 | 0.3201 | 101.89 |
| ✓ | ✗ | ✓ | 0.8517* | 0.3452 | 155.01 |
| ✓ | ✓ | ✗ | 0.6236 | 0.2411* | 52.51 |
| ✓ | ✓ | ✓ | 0.7831 | 0.2905 | 51.43* |

(b)

Table 1: **Quantitative results of conditional interpolation.** Quantitative results where the best performance is marked as (*) and the worst is marked as red. (a) Performance on CIFAR-10 and LAION-Aesthetics. AID-O and AID-I both show significant improvement over the Text Embedding Interpolation (TEI). Though Denoising Interpolation (DI) achieves relatively high fidelity, there is a trade-off with very bad performance on consistency (0.4295). AID-O boosts the performance in terms of consistency and fidelity while AID-I boosts the performance of smoothness; (b) Ablation studies on AID-O's components, showcase that the Beta prior enhances smoothness, attention interpolation heightens consistency, and self-attention fusion significantly elevates fidelity.

# 5 Experiments

**Configuration and Settings**. We evaluate quantitatively based on the three measures for conditional interpolation defined in Sec. 3.3 and user studies. Detailed experimental and application configurations are given in Appxs. F and G.

We use Stable Diffusion 1.4 [35] as the base model to implement our attention interpolation mechanism for quantitative evaluation. In all experiments, a $512 \times 512$ image is generated with the DDIM Scheduler [42] and DPM Scheduler [26] within 25 timesteps. Additional qualitative results using other state-of-the-art text-to-image diffusion models [30, 23, 2] are given in Appx. H.

## 5.1 Conditional Interpolation

**Protocol, Datasets & Comparison Methods.** For experiments in each dataset, we run 5 trials each with $N = 100$ iterations. In each iteration, we randomly select two conditions and generate an interpolation sequence with size $m = 7$. We report the mean of each metric of the interpolation sequences over all trials as the final result. Our proposed framework is evaluated using corpora from CIFAR-10 [22] and the LAION-Aesthetics dataset from the larger LAION-5B collection [39]. To the best of our knowledge, the only related method is the text-embedding interpolation (TEI) [49, 53, 55] (see Sec. 3.2). We also compare with Denoising Interpolation (DI), which interpolates along the denoising schedule; more details in DI are given in the Appx. F.

**Results.** We quantitatively evaluate our methods based on the evaluation protocol as shown in Tab. 1. AID-O significantly increases the performance of all the evaluation metrics. AID-I achieves higher smoothness, AID-O has significant improvements in consistency (-20.3% on CIFAR-10 and -23.9% on LAION-Aesthetics) and fidelity (-66.62 on CIFAR-10 and -60.37 on LAION-Aesthetics). The fidelity of AID-I is poorer than AID-O and worse than Denoising Interpolation. However, AID-I achieves competitive qualitative results as shown by the user study.

**Ablation Study.** Tab. 1 shows ablations of the AID-O framework with CIFAR-10, focusing on three primary design elements: attention interpolation, self-attention fusion, and Beta-interpolation. Results show that attention interpolation improves consistency while Beta-interpolation contributes to improvements in smoothness and self-attention fusion to enhance image fidelity. While attention interpolation (without fusion with self-attention) with Beta-interpolation achieves the highest smoothness, it does so at the cost of fidelity. Similarly, AID without Beta interpolation achieves the strongest consistency but trades off smoothness (see Fig. 4). Fig. 4 (a) provides a qualitative comparison between different ablation settings.

**User Study.** Using Mechanical Turk, we check for human preferences on four types of text sources: 1) near objects, such as dogs and cats; 2) far objects, such as dragons and bananas; 3) scenes, such as waterfalls and sunsets; and 4) scene and object, such as a sea of flowers with a robot. This variety provides a comprehensive assessment for both concept and spatial interpolation. We conducted 320 trials in total; in each trial, an independent evaluator was asked to select their preferred interpolation result. Tab. 2 shows that our method is almost always preferred, though the preference is split across AID-I and AID-O depending on the type of text sources.

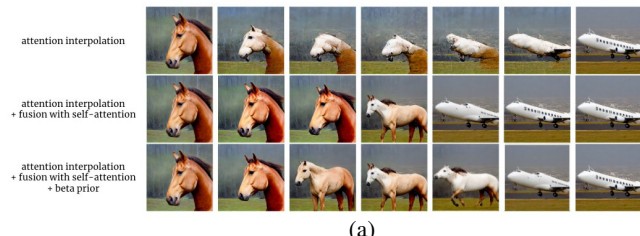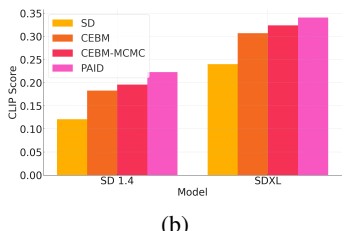

(a)                                        (b)

Figure 4: **Qualitative comparison of different ablation setting of AID.** (a) Qualitative comparison between AID without fusion (1st row), AID with fusion (2nd row), and AID with fusion and beta prior (3rd row). Fusing interpolation with self-attention alleviates the artifacts of the interpolated image significantly, while beta prior increases smoothness based on AID with fusion. (b) CLIP score of different methods on composition generation.

| Interpolation method | Near Object | Far Object | Scene | Object+Scene |
|---|---|---|---|---|
| TEI | 8.75% | 1.16% | 0% | 1.26% |
| AID-I | **53.75%** | **50%** | 45.2% | 45.57% |
| AID-O | 36.25% | 46.5% | **50%** | **51.90%** |
| Hard to determine | 1.25% | 2.32% | 4.76% | 1.26% |

| Editing Method | Smoothness |
|---|---|
| P2P | 0.3741 |
| P2P + AID | **0.8921** (0.5180 ↑) |
| EDICT | 0.5978 |
| EDICT + AID | **0.8486** (0.2508 ↑) |

(a)                                        (b)

Table 2: **Human evaluation results.** (a): Human preference ratio of each method in different categories of interpolation, AID-I, and AID-O are dominantly preferred by TEI; (b): Smoothness of different editing methods, combined with AID boosts the control ability on the editing level.

## 5.2 Application

In this section, we firstly introduce how to adapt our methods into applications including image editing control and compositional generation. We further extend to cross-modality conditions including image prompt as well with IP-Adapter [54], which enables applications including image morphing and image-controlled generation. We provide more details in Appendix. G.

**Image Editing Control**. Text-based image editing tries to modify an image based on a textual description (see Fig. 5). Existing methods [16, 11, 48] rely on text embedding interpolation to control the editing level. Training-free methods [48, 11] struggle to control the editing level based on the text, while ours does not. We validate the control ability of our methods using Prompt-to-Prompt [11] (P2P) for synthesized image editing and EDICT [48] for real image editing.

We evaluate the ability to control the editing level using the smoothness metric defined in Sec. 3.3 using the image editing dataset presented in [48]. Given an image with an editing level of 1 and the original image with an editing level of 0, we use either TEI or AID-O to interpolate edited images with levels ranging from $\{\frac{1}{6}, \frac{2}{6}, ..., \frac{5}{6}\}$ and assess the smoothness of the edited image sequence.

Quantitative results are reported in Tab. 2 (b). Our method greatly improves the smoothness of the edited image sequence, aligning with different editing levels and thereby enhancing the control ability for editing. As shown in Fig. 5, P2P alone cannot effectively control the editing level but combining it with AID allows for precise level adjustments.

**Compositional Text-to-Image Generation**. Compositional generation is highly challenging for text-to-image diffusion models [25, 6, 7]. In our experiments, we focus on concept conjunction [6] - generating images that satisfy two given text conditions. For example, given the conditions "a robot" and "a sea of flowers," the goal is to generate an image that aligns with both "a robot" and "a sea of flowers."

For compositional generation, we use PAID to interpolate between conditions $c_1$ and $c_2$ with the prompt guidance "$c_1$ AND $c_2$". For quantitative evaluation, we use the same dataset for human evaluation as in Sec. 5.1 and CLIP scores [32] to evaluate if the generated images align with both conditions. We compare our methods with vanilla Stable Diffusion [35, 30] and two other state-of-the-art training-free methods: Compositional Energy-based Model (CEBM) [25] and RRR [7].

Fig. 4 (b) shows that the CLIP score of our method is higher than previous methods for both Stable Diffusion 1.4 [35] and SDXL [30]. Moreover, our method produces fewer artifacts such as merging the two objects together, as illustrated in Fig. 6.

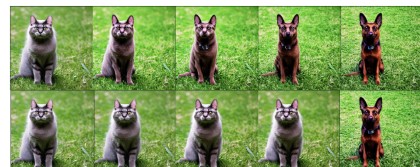 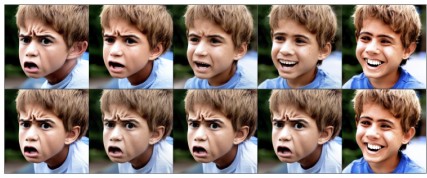

(a) "A ~~cat~~ dog sitting on the grass."          (b) "A boy is ~~angry~~ happy."

Figure 5: **Results of image editing control.** Our method boosts the controlling ability over editing. The first row of (a) and (b) is generated by P2P + AID while the second row is P2P + TEI.

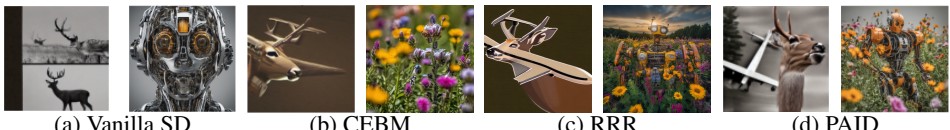

(a) Vanilla SD       (b) CEBM       (c) RRR       (d) PAID

Figure 6: **Results of compositional generation.** Images on the left are generated with "a deer" and "a plane" based on SD 1.4 [35] and images on the right are generated with "a robot" and "a sea of flowers" based on SDXL [30]. Compared to other methods, PAID-O properly captures both conditions with higher fidelity.

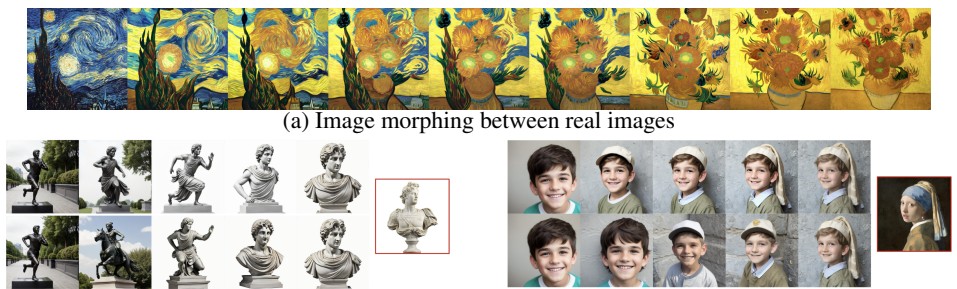

(a) Image morphing between real images

(b) "A statue is running." + global reference      (c) "A boy is smiling." + composition reference

Figure 7: **Results of AID with image conditions.** Our method is compatible with IP-Adapter for image-conditioned generation (a). In both global image prompt (b) and composition image prompt (c), from left to right the scale of additional image prompt slowly increases. The first row illustrates results controlled by AID, while the second row shows results achieved using the scale setting provided by IP-Adapter.

**Image Morphing and Image-Controlled Generation.** Image morphing finds transitions between two images, while image-controlled generation creates images based on a text prompt with an additional image prompt. To enable generation with image condition, we adapt AID on IP-Adapter [54]. IP-Adapter integrates image embeddings into cross-attention layers, allowing diffusion models to incorporate image prompts. For morphing, we use an empty text prompt and apply AID for smooth interpolation between image conditions. In image-controlled generation, AID adjusts the image prompt scale across endpoints, enhancing control.

Our method enables effective image interpolation (Fig. 7 (a)) and offers finer control than IP-Adapter. As shown in Fig. 7 (b), AID maintains both text and image alignment, while in Fig. 7 (c), it better preserves identity while following compositional references. Further comparisons are provided in Appendix G.

# 6 Conclusion

In this work, we introduce a novel task: conditional interpolation within a diffusion model, along with its evaluation metrics, which include consistency, smoothness, and fidelity. We present a novel approach, referred to as AID and PAID, designed to produce interpolations between images under varying conditions. This method significantly surpasses the baseline in performance without training, as demonstrated through both qualitative and quantitative analysis. Our method is training-free and broadens the scope of generative model interpolation, paving the way for new opportunities in various applications, such as compositional generation and image editing control.

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

# A  Preliminaries and formulation

**Linear / Spherical Interpolation**. Given tensor $A$ and tensor $B$, the linear interpolation path $r_l(w)$ where $w \in [0, 1]$ is defined as:

$$r_l(w; A, B) = (1 - w)A + wB \tag{14}$$

The spherical interpolation is defined as:

$$r_s(w; A, B) = \frac{\sin(1 - w)\theta}{\sin \theta}A + \frac{\sin w\theta}{\sin \theta}B, \quad \theta = arcos\frac{A \cdot B}{||A||||B||} \tag{15}$$

**Distinction on the Discrete Sequence and Continuous Path.** Our formulation diverges from previous studies by concentrating on the assessment of discrete samples, referred to as the interpolation sequence, instead of the continuous interpolation path. This is crucial because the quality of the interpolation sequence is determined not only by the interpolation path's quality but also by how to select the exact sample along the interpolation path, which previous methods overlook. Additionally, the size of an interpolation sequence is often low in practical usage [38, 52]. As a result, our evaluation framework is specifically designed to cater to interpolation sequences.

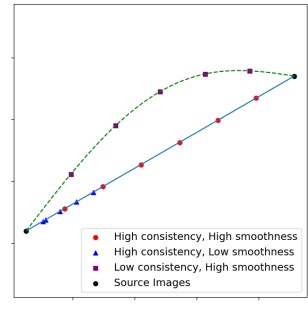

Figure 8: Difference between smoothness and consistency in measurement of discrete sequence.

This distinction is significant when evaluating smoothness and consistency as Fig. 8 shows. While Perceptual Path Length (PPL) [15] indicates both smoothness and consistency on the continuous path, where the PPL of the blue path is shorter than the green path, this does not hold in discrete sequences. The sequence can have bad smoothness even if it lies on a smooth interpolation path (see the blue triangle).

**Proof of Proposition 1**. Proposition 1 indicates that interpolating text embedding linearly is equivalent to interpolating key and value in the cross-attention mechanism. The proof is straightforward by decomposing the formula of the attention layer as follows:

$$
\begin{aligned}
A(z_i, c_i) &= Attn(Q_i, K_i, V_i) \\
&= Attn(Q_i, W_K^T c_i, W_V^T c_i) \\
&= Attn(Q_i, W_K^T r_l(\frac{i-1}{m-1}; c_1, c_m), W_V^T r_l(\frac{i-1}{m-1}; c_1, c_m)) \\
&= Attn(Q_i, r_l(\frac{i-1}{m-1}; W_K^T c_1, W_K^T c_m), r_l(\frac{i-1}{m-1}; W_V^T c_1, W_V^T c_m)) \\
&= Attn(Q_i, r_l(\frac{i-1}{m-1}; K_1, K_m), r_l(\frac{i-1}{m-1}; V_1, V_m))
\end{aligned}
\tag{16}
$$

# B  Diagnosis of Text Embedding Interpolation

**Controlled Experiments on the Key and Value of Attention.** To conduct the analysis on the key and value of self-attention and cross-attention, we analyze the effect by replacement experiments. Specifically, given two conditions $c$ and $c'$, we first generate $I$ and $I'$ accordingly. They replace all the key and value of either cross-attention or self-attention during the generation of $I'$ to the key and value computed from $I$, which incurs two new generated images including $I'_{cross}$ and $I'_{self}$. If self-attention is more important to constraint the spatial layout, The images obtained by replacing self-attention $I'_{self}$ should be more similar to $I$ compared to $I_{cross}$.

To quantitatively verify this, we consider two images sharing more similar spatial layouts should have lower differences in the low-frequency information. Therefore, we evaluate the difference in the spatial layout of the two images by directly evaluating the L2 loss on the low-pass images. Specifically, it can be written as:

$$D_{sl}(I, I'; \sigma) = \frac{1}{2}||G(I; \sigma) - G(I'; \sigma)||^2 \tag{17}$$

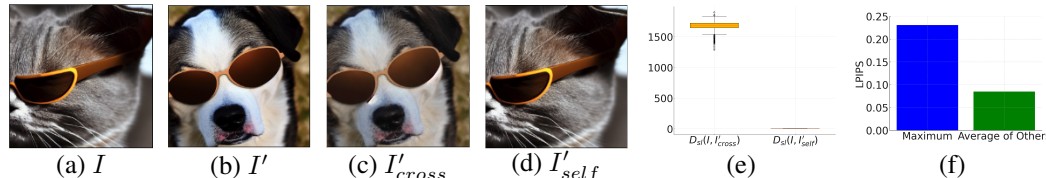

(a) $I$   (b) $I'$   (c) $I'_{cross}$   (d) $I'_{self}$   (e)   (f)

Figure 9: Diagnosis of text embedding interpolation on spatial layout (a - e) and adjacent distance (f). (a) Image generated by "a cat wearing sunglasses"; (b) Image generated by "a dog wearing sunglasses"; (c) Replacing the cross-attention during generation of (b) by (a); (d) Replacing the self-attention during generation of (b) by (a); (e) Box plot of $D_{sl}(I, I'_{cross})$ and $D_{sl}(I, I'_{self})$. When fixing a query, the key and value in self-attention mostly determine the output of pixel space compared to cross-attention. (f) The maximum adjacent distance and the average of other adjacent pairs.

where $G(\cdot; \sigma)$ represents Gaussian blurring kernel with parameter $\sigma$. We conduct our experiments based on the corpus in the form of class names of CIFAR-10 [22], which we introduce in Sec. 5.1. We run 100 trials and generate two images at each trial then compare the difference between $D_{sl}(I, I'_{self})$ and $D_{sl}(I, I'_{cross})$.

Based on our empirical verification shown in Fig. 9 (e), $D_{sl}(I, I'_{self}) \approx 0$, which indicates that the spatial layout of $I'_{self}$ is almost the same as $I$, while $D_{sl}(I, I'_{self}) >> D_{sl}(I, I'_{cross})$ indicating key and value of self-attention impulses much stronger spatial constraints on the generation than cross-attention.

**Non-smooth Distance Among Adjacent Pairs.** Selecting uniformly distributed interpolation coefficients in text embedding interpolation, commonly does not result in uniform visual transition in the pixel space. Instead, we found that small visual transitions may occur over a large range of interpolation coefficients, and vice versa. To quantitatively verify this, we randomly draw two text conditions from the same corpus of CIFAR-10 [22] and apply text embedding interpolation with uniformly distributed coefficients $\{0, 0.25, 0.5, 0.75, 1\}$ to generate interpolated images. Then we evaluate our observation by comparing the maximum distance of four adjacent pairs and the average of other distances. As Fig. 9 (f) shows, the maximum distance is often much larger than the average distance of other adjacent pairs, indicating that abrupt visual transition occurs in a short range of interpolation coefficients transition.

## C Outer vs. Inner Attention Interpolation

**Mathematical Induction**. We start by comparing the formula of outer interpolated attention and inner interpolated attention. We expand the inner interpolated attention defined in Eq. 9 as follows:

$$
\begin{aligned}
&\text{Intp-Attn}_I(Q_i, K_{1:m}, V_{1:m}; t_i) \\
=\ &\text{Attn}(Q_i,\ (1-t_i)K_1 + t_i K_m, (1-t_i)V_1 + t_i V_m) \\
=\ &\text{softmax}\left(\frac{Q_i[(1-t_i)K_1 + t_i K_m]^T}{\sqrt{d_k}}\right)[(1-t_i)V_1 + t_i V_m] \\
=\ &(1-t_i) \cdot \text{softmax}\left(\frac{Q_i[(1-t_i)K_1 + t_i K_m]^T}{\sqrt{d_k}}\right)V_1 \\
&+ t_i \cdot \text{softmax}\left(\frac{Q_i[(1-t_i)K_1 + t_i K_m]^T}{\sqrt{d_k}}\right)V_m
\end{aligned}
\tag{18}
$$

Similarly, we expand the outer interpolated attention defined in Eq. 10

$$
\begin{aligned}
&\text{Intp-Attn}_O(Q_i, K_{1:m}, V_{1:m}; t_i) \\
=\ &(1-t_i) \cdot \text{Attn}(Q_i, K_1, V_1) + t_i \cdot \text{Attn}(Q_i, K_m, V_m) \\
=\ &(1-t_i) \cdot \text{softmax}\left(\frac{Q_i K_1^T}{\sqrt{d_k}}\right)V_1 + t_i \cdot \text{softmax}\left(\frac{Q_i K_m^T}{\sqrt{d_k}}\right)V_m
\end{aligned}
\tag{19}
$$

Comparing Eq. 18 and Eq. 19 above, the essential difference is: while inner attention interpolation uses the same attention map $\text{softmax}\left(\frac{Q_i[(1-t_i)K_1 + t_i K_m]^T}{\sqrt{d_k}}\right)$ fusing source keys $K_1$ and $K_m$ for

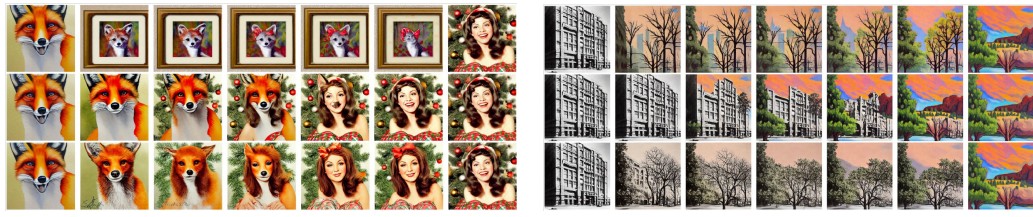

(a) *"Fox - Watercolor Art Print"* to
*"Merry-Christmas-from-AnnetteFunicello1960.jpeg"*

(b) *"Louis Henry Sullivan (September 3, 1856 – April 14, 1924) was an American architect..."* to *"Modern Landscape Painting - Zion by Johnathan Harris"*

Figure 10: **Qualitative results from LAION-Aesthetics**. For each pair of prompts, the first row is the Input Interpolation, the second row is AID-O and the third row is AID-I. Our methods provide direct and smooth interpolation in spatial layout and style, with high fidelity.

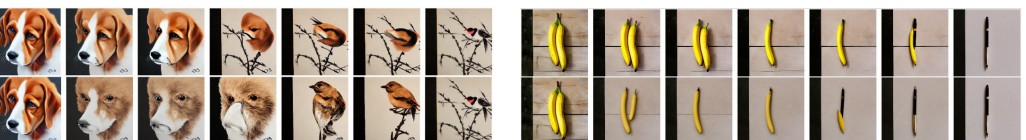

(a) *"Dog - Oil Painting"* to *"Bird - Chinese Painting"*

(b) *"banana"* to *"pen"*

Figure 11: **Qualitative comparison between AID-O (the 1st row) and AID-I (the 2nd row).** While AID-O prefers keeping the spatial layout, AID-I prefers interpolating the concept and style. Comparing the 4th column in (b), AID-I properly captures *"pen in the shape of banana"* while AID-O provides a banana but the spatial layout is the same as the pen.

different source value $V_1$ and $V_m$, outer attention interpolation, on the other hand, using different attention maps for different source key and value. This may answer why the AID-I tends to conceptual interpolation fusing the characteristics of two concepts into one target but AID-O tends to spatial layout interpolation allowing the simultaneous existence of two concepts in the interpolated image.

**Qualitative Results**. We observe that AID-I prefers interpolation on the concept or style. On the other hand, AID-O strongly enhances perceptual consistency and encourages interpolations in the spatial layout of images, as Fig. 11 shows. Even when interpolating between two very long prompts, both methods can achieve direct and smooth interpolations with high fidelity as Fig. 10 shows.

## D    Selection with Beta Prior

### D.1    Intuition behind Beta Prior

Based on our analysis in Sec. 3.4 and Sec. B, the transition often occurs abruptly in a small range of interpolation coefficients. This indicates that we need to select more points in that small range rather than uniformly select coefficients between $[0, 1]$.

We hypothesize that this is because: different from interpolation in the latent space, which is only introduced in the initial denoising steps, the diffusion model incorporates the text embedding for multiple denoising steps. This may amplify the influence of the source latent variable with higher coefficients. Therefore, when $t$ is close to 0 or 1, $r'(t)$ is closer to 0, leading to the intuition that we want to sample more mid-range $t$.

Based on our heuristics above and the empirical observation in Sec. B, we apply Beta prior, which is a bell-shaped distribution when $\alpha$ and $\beta$ are both larger than 1, to encourage more coefficients in a smaller range of interpolation coefficients. Furthermore, we can de-bias the visual transition towards one endpoint to make it smoother by adjusting $\alpha > \beta$, or vice versa.

### D.2    Dynamic Selection

The main challenge of the entire selection procedure for interpolation coefficients lies in the time cost. Firstly, the ideal interpolation coefficients can vary depending on different combinations of conditions and even different latent seeds. Secondly, exploring each new interpolation coefficient

**Algorithm 1 Exploration with Beta prior**

---

**Input:** Exploration size $n$, the initial Gaussian Noise $z_1, z_n$, two conditions $c_1, c_n$, a generative process represented as $f(z_1, z_n, c_1, c_n; w_i)$, perceptual distance function $P(\cdot, \cdot)$, CDF of Beta distribution $F_B^{(\alpha, \beta)}$
**Output:** An image sequence $\mathbf{I}$

Initialize the list of explored coefficients: $\mathbf{w} \leftarrow [0, 1]$
Initialize image sequence $\mathbf{I} = [f(z_1, z_n, c_1, c_n; 0), f(z_1, z_n, c_1, c_n; 1)]$
Initialize the list of distance of each neighbouring pair:
    $\mathbf{d} \leftarrow [P(f(z_1, z_n, c_1, c_n; 0), f(z_1, z_n, c_1, c_n; 1))]$
Initialize the hyperparameter: $\alpha \leftarrow 1$ and $\beta \leftarrow 1$, iteration number $i \leftarrow 0$

**for** $i < n$ **do**
    k = argmax($d$)               ▷ Search for the current largest distance and get its index
                         ▷ Select coefficient uniformly separate the distance under Beta prior
    $w' \leftarrow F_B^{(\alpha, \beta)^{-1}}(F_B^{(\alpha, \beta)}(\mathbf{w}_k) + F_B^{(\alpha, \beta)}(\mathbf{w}_{k+1})/2)$
                                                 ▷ Update exploration state
    Remove $\mathbf{d}_k$ from $\mathbf{d}$
    $\mathbf{d}$.insert($k, P(f(z_1, z_n, c_1, c_n; \mathbf{w}_k), f(z_1, z_n, c_1, c_n; w^prime)))$
    $\mathbf{d}$.insert($k + 1, P(f(z_1, z_n, c_1, c_n; w' f(z_1, z_n, c_1, c_n; \mathbf{w}_{k+1})))$
    $\mathbf{w}$.insert($k, w'$)
    $\mathbf{I}$.insert($(k, f(z_1, z_n, c_1, c_n; w'))$)
                                                       ▷ Update Beta prior
    Get target points: $\hat{\mathbf{w}} \leftarrow \mathbf{d}/\mathbf{d}.sum()$, $\hat{\mathbf{w}} \leftarrow$ accumulate($\hat{\mathbf{w}}$)
    Curve fit: $\alpha, \beta \leftarrow \text{argmax}_{\alpha, \beta} \text{MLE}(F_B^{(\alpha, \beta)}, \mathbf{w}, \hat{\mathbf{w}})$
    $i += 1$
**end for**
**return I**

---

requires re-running the image generation process. To address these issues, we introduce a Beta-based dynamic selection method to efficiently select satisfactory interpolation coefficients. This selection procedure is divided into two stages as shown in Alg. 1 and Alg. 2. In the first stage, we explore different coefficients based on a Beta prior and observations of perceptual distances. In the second stage, we search for a smooth image interpolation sequence from the images generated with the explored coefficients. Below, we refer to the exploration size as $n$, and the size of the interpolation sequence as $m$.

**Exploration**. During the exploration procedure, we maintain the following: 1) the currently explored coefficients $\mathbf{w}$; 2) the hyperparameters of the Beta distribution $\alpha$ and $\beta$; 3) the list of perceptual distances $\mathbf{d}$ of each neighboring image pair generated by the explored coefficients; and 4) the generated images $\mathbf{I}$.

In each iteration, we first select the neighboring image pair with the highest perceptual distance and explore a new coefficient located between the selected two coefficients according to the Beta distribution. We then generate a new image with the chosen coefficient and compute its perceptual distance with both images in the selected pair. After that, we update $\alpha$ and $\beta$ based on the new observations. Specifically, we aim to adjust the coefficients so that their differences are proportional to the perceptual distances between neighboring images, which is our target. Using the currently explored coefficients and their corresponding target coefficients, we update $\alpha$ and $\beta$ by fitting the cumulative distribution function. By repeating this process, we obtain a set of generated images to be used in the second stage.

**Search**. In the second stage, we first compute the perceptual distance between each pair (not only neighboring pairs), represented as a weight matrix $W$. We reformulate the task as a directed graph, where each image represents a node $I_i$ with $i \in \{1, 2, \ldots, n\}$, and an edge $E_{ij}$ exists for any $1 \le i < j \le n$. Our goal is to find a path starting from $I_1$ to $I_n$ with a fixed path length $m$, maximizing smoothness.

To solve this problem efficiently, we use a heuristic indicator—the difference between the maximum and minimum weights, i.e, the range of weight along the path—which reflects smoothness. This

**Algorithm 2 Search smoothest sequence**

---

**Input:** Image sequence $\mathbf{I}$, interpolation size $m$, perceptual distance $P(\cdot, \cdot)$, threshold $\epsilon$
**Output:** Smooth interpolation sequence $\mathbf{I}'$
Initialize graph $G = (V, E)$, where $V \leftarrow \mathbf{I}$ and $E_{ij} \leftarrow P(\mathbf{I}_i, \mathbf{I}_j), \forall i < j$
Set binary search bounds: $l \leftarrow 0, h \leftarrow \max(E) - \min(E)$
**while** $h - l > \epsilon$ **do**
    $D \leftarrow (h + l)/2$
    Initialize $DP$ with size $(n, m)$, where $DP_{1,1} \leftarrow (-\infty, \infty, [1])$    ▷ Each item includes (max, min, path)
    **for** $s = 1$ to $m - 1$ **do**                                         ▷ Start dynamic programming
        **for** $i = 1$ to $n - 1$ **do**
            Let $(w_{\max}, w_{\min}, \hat{I}) \leftarrow DP_{i,s}$
            **for** $j = i + 1$ to $n$ **do**
                **if** $w_{\min} < E_{ij} < w_{\max}$ **then**
                    Update $w_{\max} \leftarrow \max(w_{\max}, E_{ij})$, $w_{\min} \leftarrow \min(w_{\min}, E_{ij})$
                    **if** $w_{\max} - w_{\min} < D$ and better path found **then**
                        $DP_{j,s+1} \leftarrow (w_{\max}, w_{\min}, \hat{I} + [j])$
                    **end if**
                **end if**
            **end for**
        **end for**
    **end for**
    **if** $DP_{n,m}$ exists **then**
        $h \leftarrow D, I' \leftarrow DP_{n,m}[3]$
    **else**
        $l \leftarrow D$
    **end if**
**end while**
**return** $I'$

---

indicator is bounded by the difference between the maximum and minimum weights of the entire graph, allowing us to perform a binary search to find the lowest possible difference. Specifically, given a value of such a difference, we search for a path fulfilling the requirement with length $m$ using dynamic programming. The algorithm is guaranteed to find the path with the minimal difference between the maximum and minimum weights. This is equivalent to finding an image interpolation sequence with the minimal difference between the maximum and minimum perceptual distances among all neighboring pairs.

The computation complexity of the search algorithm is $O(n^2 m \cdot log(c))$, where $c$ is the range of the perceptual distance. In practice, we choose the exploration size $n = 1.5m$, which can already achieve very smooth results and the overhead is neglible compared to the cost of inference with diffusion model.

## E   Trade-off between Consistency and Fidelity via Warm-up Steps

We observe that early steps in denoising are essential to determine the spatial layout of the generated image. Thus, we can trade off between the effect of interpolation and prompt guidance by setting the number of warm-up steps. After several warm-up steps, we transform the attention interpolation into a simple generation process.

This design is based on the observation that early denoising steps of the generative model can determine the image content to a large extent as Fig. 12 shows. With only 5 initial steps (over a total of 25 denoising steps) using "dog" as guidance (the 6th image in Fig. 12, the image content is already fixed as "dog", which means the influence of later denoising steps using "car" has very low influence to the image content generation.

Therefore, we can utilize this characteristic of the diffusion model to constrain spatial layout with AID in the early stage of denoising and then transit to self-generation with the guided prompt to refine the details.

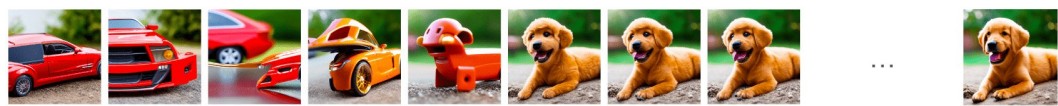

Figure 12: **Effect of early denoising steps.** The images are generated using 25 denoising steps. For the $i$th image shown in the row from left to right, it is generated by using *"A photo of dog, best quality, extremely detailed"* in the first $i - 1$ denoising steps, then generated by using *"A photo of car, best quality, extremely detailed"* for the rest denoising steps.

Figure 13: Screenshot of the survey layout. The user is prompted to choose the best interpolation sequence with high smoothness, consistency, and fidelity.

# F  Auxiliary Experiments Details

**Hardware Environments.** All quantitative and qualitative experiments presented in this work are conducted on a single H100 GPU and Float16 precision.

**Perceptual Model Used in Evaluation Metrics.** For consistency and smoothness, we follow conventional settings and choose VGG16 [41] to compute LPIPS [57]. For fidelity, we adapt the Google v3 Inception Model [44] following previous literature to compute FID between source images and interpolated images.

**Datasets.** We introduce the details of CIFAR-10 and LAION-Aesthetics used for evaluating conditional interpolation here.

- **CIFAR-10**: The CIFAR-10 dataset [22] comprises 60,000 32x32 color images distributed across 10 classes. This dataset is commonly used to benchmark classification algorithms. In our context, we utilize the class names as prompts to generate images corresponding to specific categories. The CIFAR-10 corpus aids in assessing the effectiveness of our framework, PAID, in handling brief prompts that describe clear-cut concepts.
- **LAION-Aesthetics**: We sample the LAION-Aesthetics dataset from the larger LAION-5B collection [39] with aesthetics score over 6, curated for its high visual quality. Unlike CIFAR-10, this dataset provides extensive ground truth captions for images, encompassing lengthy and less direct descriptions. These characteristics present more complex challenges for text-based analysis. We employ the dataset to test our framework's interpolation capabilities in more demanding scenarios.

**Selection Configuration.** In terms of Bayesian optimization on $\alpha$ and $\beta$ in the beta prior to applying our selection approach, we set the smoothness of the interpolation sequence as the objective target, $[1, 15]$ as the range of both hyperparameters, 9 fixed exploration where $\alpha$ and $\beta$ are chosen from $\{10, 12, 14\}$, and 15 iterations to optimize.

**Denoising Interpolation.** Denoising interpolation interpolates the images along the schedule. Specifically, given prompt A and prompt B and the number of denoising steps $N$, for an interpolation coefficient $t$ we guide the generation with prompt A for the first $\lfloor tN \rfloor$ steps and guide with prompt B for the rest of steps.

**Human Evaluation Details.** To minimize bias towards a particular style, we included an equal number of photorealistic and artistic prompts for each category. We conducted 320 trials in total. In each trial, an independent rater from Amazon Mechanical Turk evaluated the results and chose the best one among AID-I, AID-O, and text embedding interpolation (TEI). We show the layout of the human study survey in Fig. 13. For near object, the prompt is sampled from: {["a dog", "a cat"], ["a jeep", "a sports car"], ["a lion", "a tiger"], ["a boy with blone hair", "a boy with black hair"]};

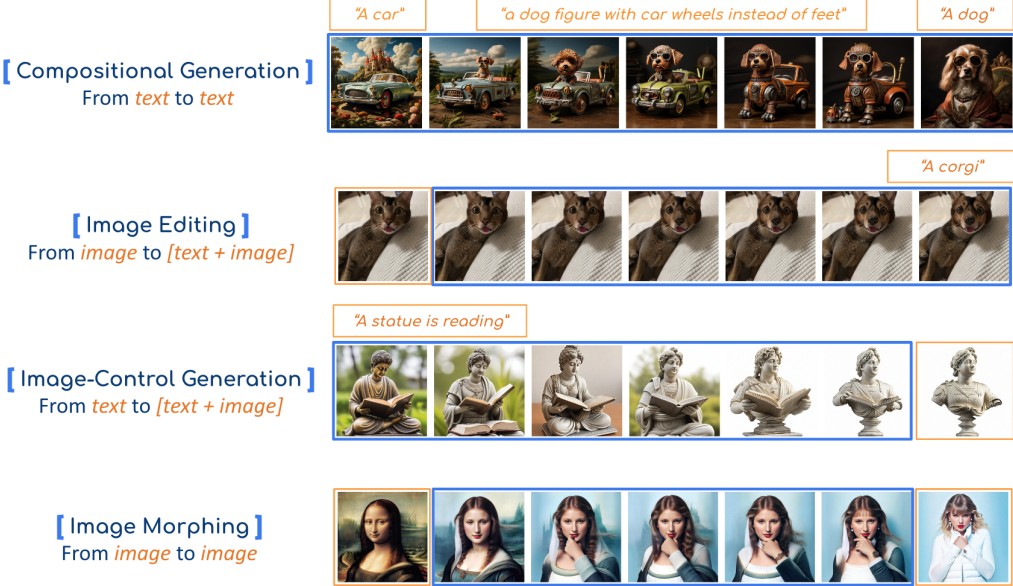

Figure 14: Our method combined with the inversion method [48] or IP-Adapter [54] can be further applied to several downstream tasks including image editing, image-control generation and image morphing.

For far object, the prompt is sampled from: {["an astronaut", "a horse"], ["a girl", "a ballon"], ["a dragon", "a banana"], ["a computer", "a ship"], ["a deer", "an airplane"]}; For scene, the prompt is sampled from: {["sunset", "moonlit night"], ['moonlit night', 'forest'], ["forest", "lake"], ['lake', 'sunset']}; For scene and object, the prompt is sampled from: {["a robot", "sea of flowers"], ["a deer", "urban street"], ["sea of flowers", "a deer"], ["urban street", "a robot"]}.

# G  Details of Applications

In this section, we introduce to adapt our methods for four applications including **composition generation** (interpolation between text conditions), **image editing** (interpolation from image condition to multi-modal condition), **image morphing** (interpolation between image conditions) and **image-conditioned generation** (interpolation from text condition to multi-modal condition), as Fig. 14 shows, where the content in orange box represents the input, and the content in blue box represents the output. We combine our method with IP-Adapter [54] for the application of image-control generation and image morphing.

## G.1  Composition Generation: Text to Text

**CEBM [25]** interprets diffusion models as energy-based models in which the data distributions defined by the energy functions can be combined, and generate compositional images by considering the generation from each condition separately and combining them at each denoising step.

**RRR [7]** concludes that the sampler (not the model) is responsible for the failure in compositional generation and proposes new samplers, inspired by MCMC, which enables successful compositional generation, with an energy-based parameterization of diffusion models which enables Metropolis-corrected samplers.

**Datasets.** We use the same dataset for human evaluation introduced in Sec. F.

## G.2  Image Editing Control: Image to {Text + Image}

**P2P [11].** controls the prompt-to-prompt editing, i.e., editing on synthesized images while keeping spatial layout by borrowing the cross-attention map, i.e., query and key, from the generation of the edited image. P2P enables editing the focus content while keeping the spatial layout the same for synthesized images.

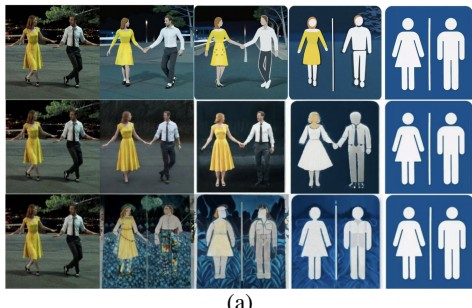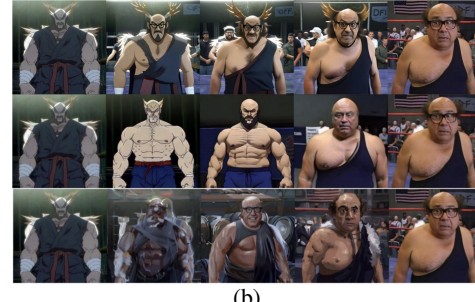

|(a)|(b)|

Figure 15: **Results of image morphing.** We compare our method (first row) with IMPUS [53] (second row) and the method by Wang et al. [49] (third row). Our approach achieves comparable or superior performance to IMPUS, but with only 1/100th of the time cost.

**P2P + AID.** For a given source prompt generating source images, we view another generation trajectory of edited images as a whole and apply AID on the two-generation trajectory. Specifically, with P2P, the interpolation of AID between cross-attention only happens at the value vector while other components remain the same as the original method.

**EDICT [48].** re-conceptualizes the denoising process through a series of coupled transformation layers, with each inversion process mirrored as such transformations. We denote the comprehensive process in EDICT of generating image $I$ from latent $z$ under prompt $C$ as $E_f(z, C)$, and its inverse as $z = E_i(I, C)$. AID is applied within these coupled layers during the denoising phase. We explore two applications: editing control and video frame interpolation.

**EDICT + AID.** For a given image $I_1$ with source prompt $C_1$ and target prompt $C_t$, we first derive its latent representation $z_1 = E_i(I_1, C_1)$ by EDICT. To interpolate $m$ images between $C_1$ and $C_t$, we replicate $z_1$ across $z_{1:m}$ and employ AID for sequence generation.

**Dataset.** We follow the same dataset presented in [48] for quantitative evaluation. For synthesized images, each data is presented as a source prompt and an editing prompt. For real images, each data is presented in a source image and editing prompt. Specifically, images of five classes "African Elephant, Ram, Egyptian Cat, Brown Bear, and Norfolk Terrier" from ImageNet [4] are taken and then we conduct four types of experiments: one involves editing "a photo of {animal 1}" to "a photo of {animal 2}" (resulting in 20 species editing pairs in total); two involve contextual changes ("a photo of {animal} in the snow" and "a photo of {animal} in a parking lot"); and one involves a stylistic change ("an impressionistic painting of the {animal}"). When this is applied to synthesized images, we use "a photo of the {animal}" as the source prompt.

### G.3 Image Morphing: Image to Image

Image morphing involves smoothly transitioning from one image to another by blending and aligning key features. Traditional approaches often require fine-tuning pretrained text-to-image diffusion models on individual samples [53, 55], which can be computationally intensive and time-consuming. In contrast, our method provides a training-free solution, allowing seamless image transitions without the need for fine-tuning. We compare our approach with [51, 53].

**AID for Image Morphing**. We build on IP-Adapter [54], which adapts text-to-image diffusion models to accommodate multi-modal conditions, including image and text inputs. IP-Adapter manages image prompts through cross-attention layers, enabling our method to naturally extend to image morphing without additional training. For each morphing task, the provided image serves as the prompt in IP-Adapter, with a null text input. Unlike approaches that rely on model fine-tuning [55, 53], our method operates entirely training-free, making it a more efficient alternative while delivering high-quality results.

**Preliminary Results**. As shown in the first row of Fig. 15, our method produces smooth, consistent transitions for real-world images. In comparison to the previous training-free approach by [49] (third row in Fig. 15), our method achieves improved fidelity in the interpolated images. Furthermore, compared to IMPUS [53] (second row in Fig. 15), which requires fine-tuning Stable Diffusion with LoRA [14]—taking approximately one hour per sample on a single A100 GPU—our approach

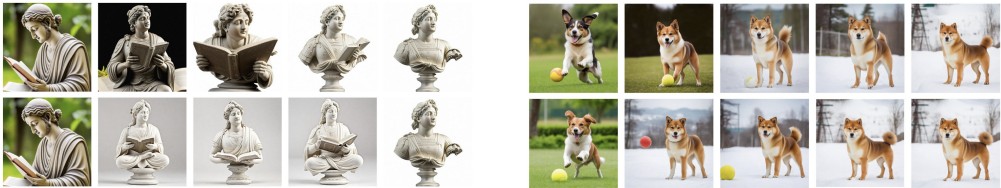

(a) "A statue is reading a book"  (b) "A dog is playing with a ball"

Figure 16: **Results of image-conditioned generation.** IP-Adapter (second row) has difficulty properly scaling the influence of the additional image condition (see the last column in (a) and (b)). Our method (first row) achieves smoother control and greater subject consistency, particularly evident in the statue's hair in (a).

is training-free, generating the entire interpolation sequence in about six minutes, achieving competitive performance with significantly lower computational cost.

### G.4 Image-Conditioned Generation: Text to {Text + Image}

Image-conditioned generation has emerged as an approach for guiding text-to-image models using supplementary control signals that are challenging to express through text alone, such as layout, subject, and style. Existing methods, including ControlNet [56] and IP-Adapter [54], often struggle to balance these additional conditions effectively, particularly when scaling their influence. Our method addresses these limitations, specifically improving upon IP-Adapter, which frequently fails to maintain a balance between the image and text inputs when both are used as conditions.

**AID for Image-Conditioned Generation**. Our approach utilizes the IP-Adapter integrated with the AID framework, similarly to its use in image morphing. For image-conditioned generation, however, we begin with a null image prompt alongside a text prompt, and at the interpolation endpoint, we incorporate both the image and text prompts.

**Preliminary Results**. Fig. 16 compares our method (first row) with IP-Adapter (second row). The leftmost images are generated without the additional image condition, while the rightmost images show the maximum influence of the image condition. IP-Adapter fails to maintain subject consistency, as shown by the statue's inconsistent hair in (a), and struggles to scale the image condition effectively, especially in (b). In contrast, our method offers more consistent and smoother control over the subject, ensuring a better balance between image and text conditions.

## H Auxiliary Qualitative Results

We show more qualitative results here using prompt guidance with inner attention interpolation. In this section, the results are obtained with Stable Diffusion 1.5 [35] and UniPCMultistepScheduler [58]. To enhance the visual ability, we use the negative prompt "monochrome, lowres, bad anatomy, worst quality, low quality". To trade-off between perceptual consistency and effectiveness of prompt guidance, we use the first 10 denoising steps over 50 total denoising steps of Uni for warming up. As Fig. 17 and Fig. 18 show, our methods can generate image interpolation on different concepts and paintings. We provide more examples in Fig. 19. And we provide more results obtained by SDXL [30] and Animagine 3.0 [23] in Fig. 20.

## I Distinction with Concurrent Works on Image Morphing

There are two concurrent works [55, 53] that also focus on deep interpolation but the objective is different where their objective is on real-world image morphing where the main challenge is to make the interpolation between real images is as good as the interpolation between generated images. On the contrary, we focus on improving the quality of generative interpolation, which can be further used in their framework.

**IMPUS [53]** specializes in generating image morphing through uniform perceptual sampling. The process begins with applying textual inversion to derive text embeddings, followed by fine-tuning the model with LoRA [14] based on specific image content. The generation of image sequences is executed sequentially, ensuring the images are uniformly distributed.

**DiffMorpher [27]** initiates its process by training with LoRA, utilizing both prompts and source images. The method extends beyond simple interpolation of text embedding and latent space by also interpolating within LoRA parameters. They also explore attention interpolation, specifically within the realm of inner attention. They observe that attention interpolation across all denoising steps can introduce artifacts, which makes them adopt it with interpolating with other multiple components. On the contrary, we find that combining outer attention interpolation with self-attention fusing can significantly address this problem, and the performance is boosted without any fine-tuning, which emphasizes the difference.

IMPUS and DiffMorpher lack control over the specific interpolation path due to their reliance on interpolated text embeddings. Conversely, our method can plug in to allow precise control over the interpolation path.

Furthermore, IMPUS and DiffMorpher necessitate fine-tuning during the testing phase to achieve optimal performance to tackle the challenges from real images, requiring thousands of iterations to optimize LoRA or text embeddings for each interpolation sequence. Our method is more efficient for downstream tasks such as editing and video frame interpolation in a training-free manner.

## J   Limitation and Social Impact

**Limitation.** Our method is post-hoc performed on a text-to-image diffusion model and the results are dependent on the ability of the base model.

**Social Impact.** Our method offers control over training-free image editing methods which initially have nearly no such ability. This is impactful to the practical usage of the text-to-image diffusion model. However, our method also increases the compositional generation ability of the text-to-image model, which may make deepfake harder to detect.

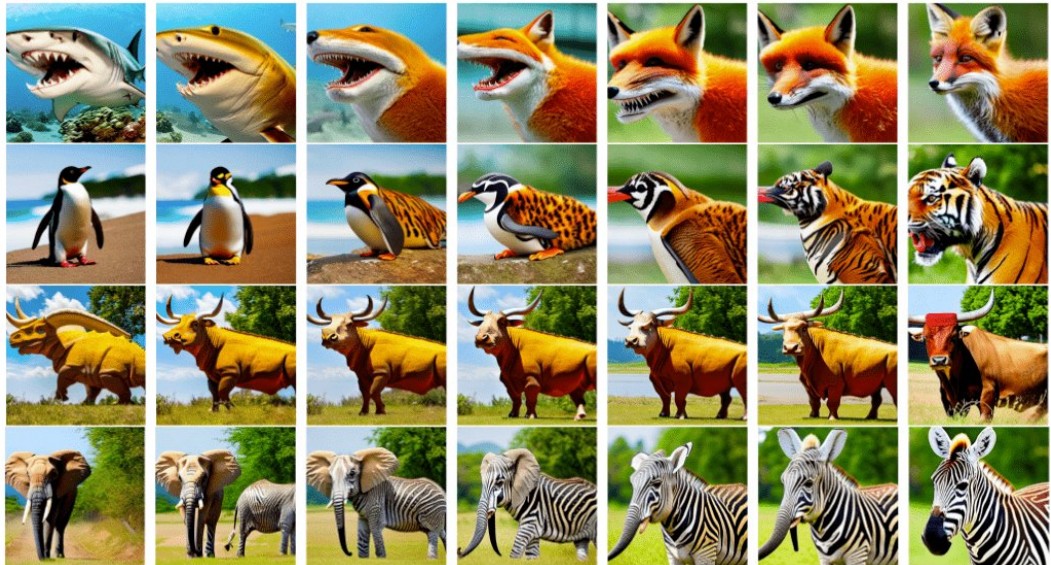

Figure 17: **Qualitative results of interpolation between animal concepts**. For an animal, we use "A photo of $\{animal\_name\}$, high quality, extremely detailed" to generate the corresponding source images. The guidance prompt is formulated as "A photo of an animal called $\{animal\_name\_A\}$-$\{animal\_name\_B\}$, high quality, extremely detailed". PAID enables a strong ability to create compositional objects.

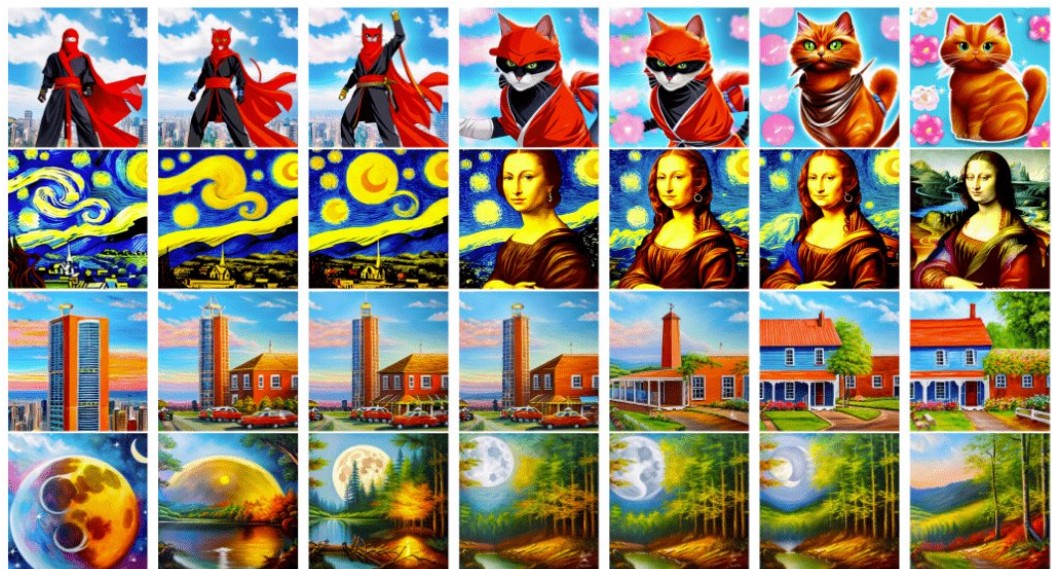

Figure 18: **Qualitative results of interpolation between different paintings.** For a painting, we use "A painting of $painting\_name$, high quality, extremely detailed" to generate the source images. The guided prompt is generated by GPT-4 [28] given description of source images, e.g., the guided prompt for the second row is "A painting of Mona Lisa under Starry Night, high quality, extremely detailed".

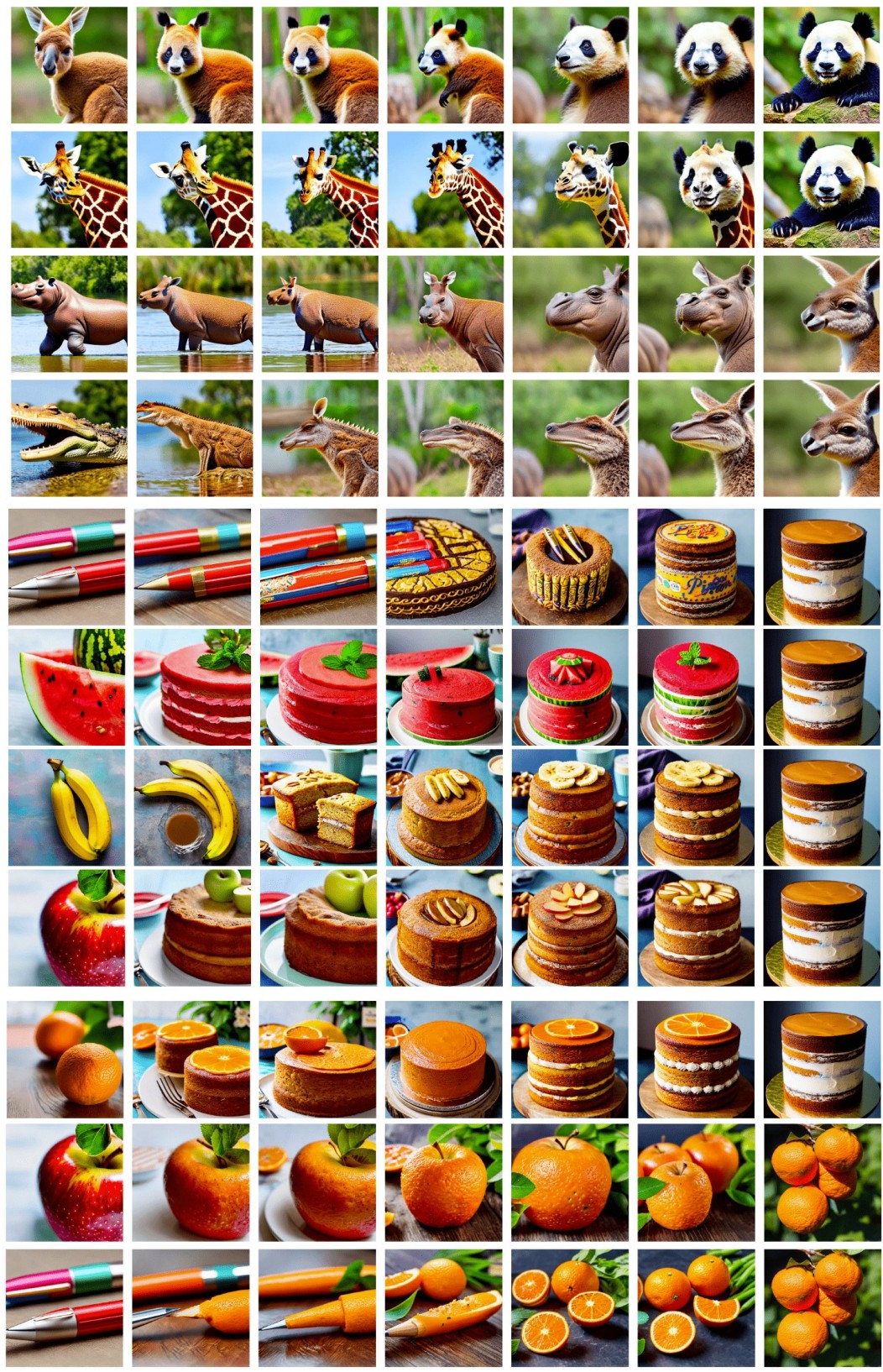

Figure 19: **More qualitative results generated by SD 1.5.**

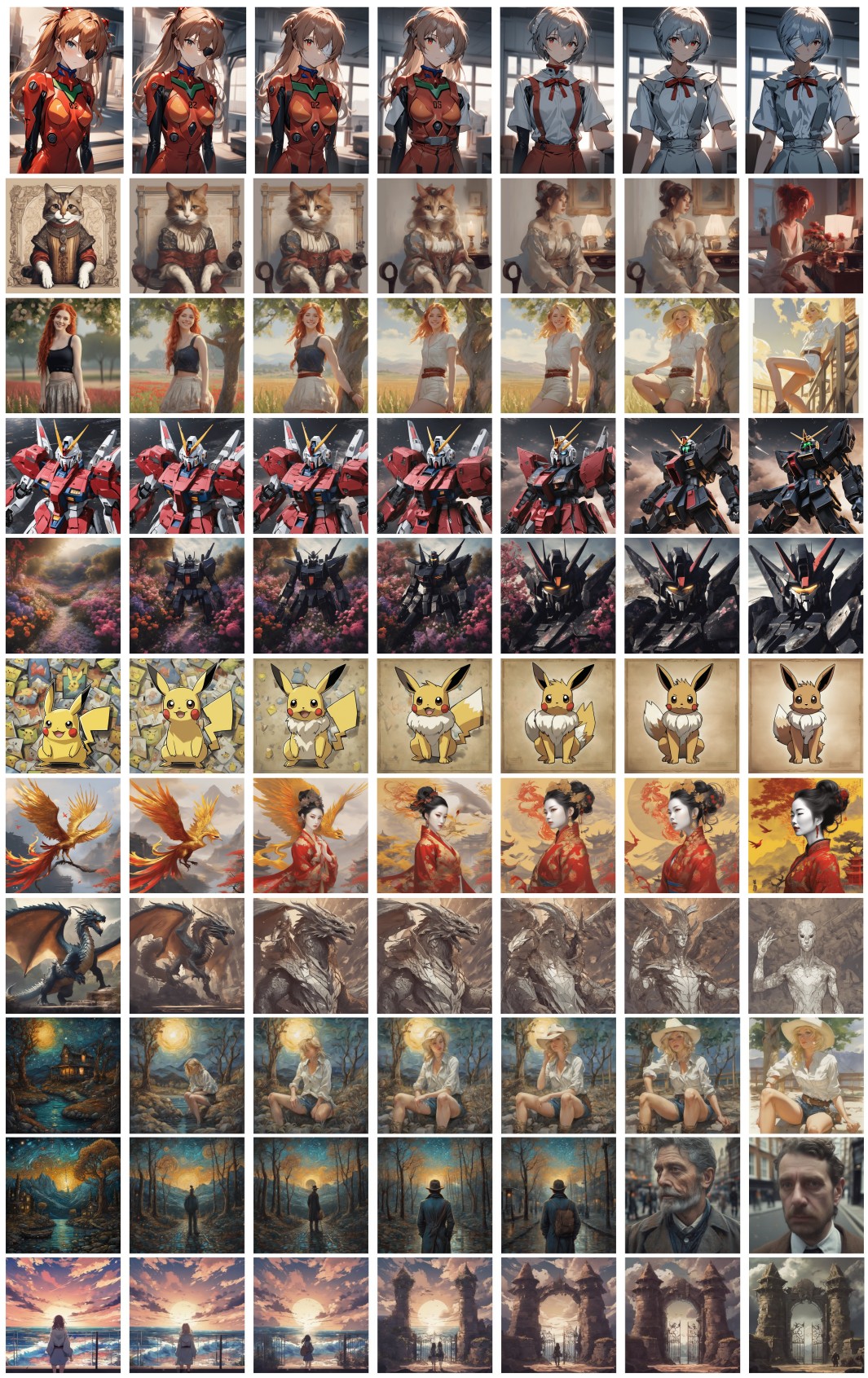

Figure 20: **More qualitative results generated by Animagine 3.0 [23] (the 1st row) and SDXL (from 2nd to 9th rows).**

