# OpenReview forum: "AID: Attention Interpolation of Text-to-Image Diffusion"
_NeurIPS.cc/2024/Conference — NeurIPS 2024 poster_

### Official Review · Reviewer_frah · 2024-06-28

**Soundness:** 3
**Presentation:** 3
**Contribution:** 3
**Rating:** 4
**Confidence:** 4

**Summary:**

This paper proposes a training-free method for generation interpolation of diffusion models with attention manipulation. Targeting on the layout transition inconsistency and nonuniform step-wise transition, the paper proposes to extend the attention interpolation from previous cross-attention to self-attention, and adopts a beta distribution to ease the nonuniform transition. Additionally, three metrics are proposed to evaluate the interpolation qualities, involves consistency, smoothness and fidelity. Experiments show the effectiveness.

**Strengths:**

1. The beta distribution for interpolation weights instead of linear weights seems promising.
2. The proposed metrics seem sound to evaluate the interpolation quality.
3. Experiments show the effectiveness of the proposed attention interpolation.

**Weaknesses:**

1. The application of the generation interpolation seems somewhat restricted, and the practical meaningness of the task is doubted, is the interpolation could benefit other generative applications, or isolated.
2. Basically, the manipulation of the attention map has been broadly explored for image editing, such as [Masactrl; ICCV 2023], [InfEdit; CVPR 2024], etc., and the self-attention corresponding to the layout and the cross-attention corresponding to the semantic are basically common sense, the proposed AID adopts a similar fashion with reformulated interpolation task, where the contribution is somewhat insignificant.
3. The discussion with training-based interpolation methods is suggested to be provided, such as [UniHDA; 2024].

**Questions:**

1. What's the behavior difference between the inner-interpolation and outer-interpolation, is there any guide lines when to use the inner or outer?

**Limitations:**

The authors have adequately discuss the limitations and broader impacts of their work.

---

> ### Author Rebuttal · Authors · 2024-08-06
>
> We thank Reviewer frah for the feedback. Here, we aim to clarify our contributions and the application of the proposed methods.
>
> > W1: The application of the generation interpolation seems somewhat restricted, and the practical meaningness of the task is doubted, is the interpolation could benefit other generative applications, or isolated.
>
> We respectfully disagree with the assertion of limited applications. On the contrary, we believe the applications we mentioned are of significant importance. As discussed in Sections 5.2 and 5.3, the proposed method enhances performance in two critical areas: image editing control and compositional generation. These applications, and the value of exploring interpolation, have been recognized for their value in numerous existing works accepted in the top conference [6,7,11,25,38,49,50].
>
> To the best of our knowledge, our method is the first to efficiently adjust the editing level, particularly in a tuning-free manner. Traditional editing methods are incapable of this, making our approach more practical for real-world applications, as demonstrated in Section 5.2.
>
> Furthermore, our method can be independently used for compositional generation in a tuning-free manner, outperforming previous state-of-the-art methods as shown in Section 5.3. It is noteworthy that even state-of-the-art models [2,30,35] struggle with generating two-concept images. Our method plays a crucial role in addressing this challenge and improving performance.
>
> > W2: Basically, the manipulation of the attention map has been broadly explored for image editing, such as [Masactrl; ICCV 2023], [InfEdit; CVPR 2024], etc., and the self-attention corresponding to the layout and the cross-attention corresponding to the semantic are basically common sense, the proposed AID adopts a similar fashion with reformulated interpolation task, where the contribution is somewhat insignificant.
>
> We respectfully disagree with the assertion of insignificant distribution.
>
> As acknowledged in Section 2, attention manipulation is extensively explored [1,3,11,31,34,51], particularly from a downstream application perspective such as image editing and compositional generation. The trade-off between edits and retaining parts of the original image is crucial for practical applications in image editing, as highlighted in [2,16]. This issue is also a focal point in compositional generation research [6,7,25].
>
> However, existing works recognize the trade-off issue as an open challenge without effectively addressing it. When attempting such trade-offs, they often rely on varying scales of classifier-free guidance and simple text embedding interpolation [16,49,57], which we demonstrate to be very ineffective and lacking generalization ability in Sections 3.4 and 5. [Masactrl; ICCV 2023] and [InfEdit; CVPR 2024] also lack this trade-off capability, and we will include additional results of more attention-based methods even for training-based methods in our revision. Given the extensive use of attention mechanisms, our work provides practical insights on controlling the level of attention manipulation, assisting existing works and addressing a critical gap in current research.
>
> > W3: The discussion with training-based interpolation methods is suggested to be provided, such as [UniHDA; 2024].
>
> Our method diverse from training-based interpolation methods [58,59] in mainly two aspects. The main advantage of training-based interpolation methods is that these kind of work often focus on cross-model / cross-domain area, enabling interpolation across domain. However, they often requires sample-wise fine-tuning to adapt the source generator, serving as the main challenge to real world application. On the contrary, our method only focus interpolating within text modality, but it is tuning-free and more efficient while keeping the competitive performance.
>
> We kindly thanks for the advice, and will add more discussion with [59] together in the revision.
>
> > Q1: What's the behavior difference between the inner-interpolation and outer-interpolation, is there any guidelines when to use the inner or outer?
>
> Thanks for raising this point.  Generally, we recommend to use outer interpolation (AID-O) when the interpolation is more related to spatial layouts, and (inner interpolation) when interpolating between distinct semantic concepts.  A discussion is provided in Appendix C and Fig. 10.
>
> Additional References:
>
> [57] Qixun Wang, Xu Bai, Haofan Wang, Zekui Qin, Anthony Chen, Huaxia Li, Xu Tang, and Yao Hu. Instantid: Zero-shot identity-preserving generation in seconds, 2024.
>
> [58] Rinon Gal, Or Patashnik, Haggai Maron, Gal Chechik, and Daniel Cohen-Or. Stylegan-nada: Clip-guided domain adaptation of image generators, 2021.
>
> [58] Hengjia Li, Yang Liu, Yuqi Lin, Zhanwei Zhang, Yibo Zhao, weihang Pan, Tu Zheng, Zheng Yang, Yuchun Jiang, Boxi Wu, and Deng Cai. Unihda: A unified and versatile framework for multi-modal hybrid domain adaptation, 2024.
>
> Other references are notated as the same as the main paper.

---

> > ### Comment · Reviewer_frah · 2024-08-14
> > **Official Comment by Reviewer frah**
> >
> > Thanks for the authors rebuttal. Basically, the mentioned image editing and compositional generation refer to the feature interpolation and prompt interpolation, and the proposed AID could provide the strength adjustmnet, which may still somewhat limited in application, and what the user can control is quite less. As the main concern is still exist, I maintain my score.

---

> > > ### Author Response · Authors · 2024-08-14
> > > **Clarification on applicability**
> > >
> > > Thank you for your feedback.
> > >
> > > We would like to clarify that the proposed AID does not solely provide strength adjustment; it also offers guidance for the interpolation path (PAID), as detailed in Figure 1(f) and Section 4.3.
> > >
> > > Secondly, the image editing and compositional generation referred do not equate to feature interpolation and prompt interpolation. These are technical challenges recognized by many papers accepted at NeurIPS, as mentioned in our previous response. Our method can enhance many existing works that rely on attention manipulation on image editing, and it can be independently applied to compositional generation.
> > >
> > > Thirdly, we want to emphasize that our method is applied independently to compositional generation, and it outperforms existing state-of-the-art methods. Numerous papers focused solely on addressing this issue have been accepted at NeurIPS and other top conferences, and they are not considered "limited" in their applications [6,7,25].
> > >
> > > Furthermore, our objective of the main paper is not to deliver a product but to investigate an under-explored technical problem. And the conditional interpolation problem itself, is interesting and important, which is recognized by other reviewers.
> > >
> > > We hope that this could address the concern on the applicability espeicially considering the state-of-the-art performance on compositional generation individually.

---

### Official Review · Reviewer_THPg · 2024-07-10

**Soundness:** 2
**Presentation:** 3
**Contribution:** 3
**Rating:** 5
**Confidence:** 4

**Summary:**

Summary and Contributions: The authors introduce a novel task called conditional interpolation, which is to generate interpolation images with various conditions like text or pose. They propose an attention-base (and prompt-guided) method to achieve conditional interpolation, and three evaluation metrics to assess the consistency, smoothness, and fidelity of generated images. Extensive experiments shows their method achieve the best performance in image interpolations of diffusion models.

**Strengths:**

1. The propose task(Conditional interpolation) is unexplored and interesting. Comom image interpolation task focus on generate transition images between two real-world images, but limit to one condition. The author introduce both text-embedding and attention mechanism to achieve better performance.
2. The authors do a detail analysis of conditional interpolation, specifically text-embedding interpolation. They prove that text-embedding interpolation is equivalent to manipulating the keys and values in cross-attention module and find that doing similar operation in self-attention layer can significantly improve spatial consistency.
3. The method can be used in image interpolation with distinct conditions like “a truck” and “a cat”. The authors also show their method can improve image-editing results when use editing methods like p2p.
4. The authors introduce a third prompt to guide the interpolation between two prompts, which is interesting and useful.

Correctness:
the claims and method are correct.
Clarity:
The paper is well written
Relation to Prior Work:
The paper is clearly discussed how their work differs from previous contributions.

**Weaknesses:**

1.The paper lack of a clear and detail definition of the conditional interpolation. The authors claims that they formulate a new task call conditional interpolation, which is doing interpolation under various condition, such as text and pose. But I am confused that the method they proposed only condition on text, how could it be various condition? What the definition about various condition?
2.The comparations between baselines are inconsistent. The table 1 shows the result of TEI and DI, but the table 2 only show TEI.
3.The qualitatively compare between baselines（TEI and DI）is lack, while the quantitative result of this two baseline are exist.

**Questions:**

No coded is provided, which I think is important.

**Limitations:**

see weaknesses.

---

> ### Author Rebuttal · Authors · 2024-08-06
>
> We thank Reviewer THPg for the constructive feedback. Here is the response to reviewer's concerns.
>
> > W1:The paper lack of a clear and detail definition of the conditional interpolation. The authors claims that they formulate a new task call conditional interpolation, which is doing interpolation under various condition, such as text and pose. But I am confused that the method they proposed only condition on text, how could it be various condition? What the definition about various condition?
>
> Thanks for raising this point. We state in the title and introduction that this work targets text-to-image diffusion models where text serves as condition.  We will remove mentions of ``various'' conditions.  Note we give a clear definition of the task formulation and the evaluation metrics in Sec. 3.2. We are happy to further highlight these as definitions in the revision.
>
> > W2:The comparisons between baselines are inconsistent. Table 1 shows the result of TEI and DI, but table 2 only shows TEI. 3.The qualitative comparison between baselines (TEI and DI) is lacking, while the quantitative results of these two baselines exist.
>
> Thanks for pointing this out. We are happy to add some results to the revision as reference for our own work.
>
> Note we omitted DI for the qualitative results and human study because it is not used by competing methods; there is no comparison basis for Table 2 and the qualitative results.
>
> > Q1:No coded is provided, which I think is important.
>
> Our intention was to release the source code upon acceptance. We provide an anonymous link to our code in a seperate comment to AC following the rebuttal rule of NeurIPS.

---

### Official Review · Reviewer_oFS1 · 2024-07-13

**Soundness:** 2
**Presentation:** 3
**Contribution:** 2
**Rating:** 4
**Confidence:** 3

**Summary:**

In this work, the authors propose Attention Interpolation via Diffusion (AID), a novel, training-free technique for improving image interpolation under specific conditions like text or pose. Traditional methods using linear interpolation often produce inconsistent, low-fidelity images. AID enhances image consistency and fidelity with a fused interpolated attention layer and selects interpolation coefficients using a beta distribution for smoother results. An advanced variant, Prompt-guided Attention Interpolation via Diffusion (PAID), treats interpolation as a condition-dependent generative process. The authors include the experiments to demonstrate AID's consistency, smoothness, and efficiency in condition-based interpolation. The work also includes user study to show AID better aligns with human preferences and aids compositional generation and image editing control.

**Strengths:**

1. The work includes user study to compare performance of different models which is appreciated.
2. The work is motivated to improve existing image interpolation methods based on their drawbacks.

**Weaknesses:**

1. The work proposes a few metrics to evaluate the quality of interpolated images. However, these metrics could be biased and fail to evaluate the quality of samples. More details are discussed in Questions.
2. Though image interpolation can generate interesting visual effects. I feel the actual applications for such technique could be limited.

**Questions:**

1. For the fidelity metric in Eq. 7, one should expect the metric is maximized when the interpolation starts the same as image A, has an abrupt change from image A to B, and stays as image B for the remaining. This could be against the objective to generate perceptually consistent and smooth images.
2. Based on proposition 1, text embedding interpolation can be viewed as manipulating the K/V which leads to suboptimal interpolation. However in (fused) inner-interpolation, still only K/V are manipulated.
3. Is there a systematic way to determine the coefficient for Beta distribution, or is it a hyperparameter that needs tune for different models or prompts?
4. How does applying interpolation module in self-attention or cross-attention alone affect the performance?

**Limitations:**

The work sufficiently addressed the limitations.

---

> ### Author Rebuttal · Authors · 2024-08-06
>
> We thank reviewer oFS1 for the feedback. We provide our response to the reviewer's concerns in the following.
>
> > W1 (Q1): For the fidelity metric in Eq. 7, one should expect the metric is maximized when the interpolation starts the same as image A, has an abrupt change from image A to B, and stays as image B for the remaining. This could be against the objective to generate perceptually consistent and smooth images.
>
> Thanks for pointing this out. Our fidelity metric Eq. 7 is based on the Fréchet Inception Distance (FID).  As an aside, Eq. 7 is based on $n$ interpolated sequences in the dataset and not a single sequence.  Nonetheless, duplicating the $n$ source image pairs as the interpolations will lead to the max FID.
>
> However, such a degenerate setting is true of the FID measure in general and have been discussed in existing works [57,58]. \emph{Any} generative model achieves the highest FID when it duplicates the reference image set.  Yet FID is widely accepted in the literature to evaluate fidelity for image generation [2,30,35] and image interpolation [38,50].  This is because existing works do not evaluate with FID alone; instead, good models must have high fidelty \emph{and} diversity. In our case, we evaluate with fidelity, consistency, and smoothness.
>
> Given the pervasive use of FID for evaluating fidelity, we believe our adoption of FID should not be considered a significant weakness. We are happy to add discussion in the revision about the importance of achieving strong performance on a diverse set of measures.
>
> > W2: Though image interpolation can generate interesting visual effects. I feel the actual applications for such technique could be limited.
>
> We respectfully disagree. Sections 5.2 and 5.3 already show our method's effectiveness in two significant applications: image editing control and compositional generation.
>
> Image editing remains a prominent research topic over the past two years [11,49,59]. A main challenge is controlling the level of editing and Sec. 5.2 demonstrates that existing editing methods often fail. On the contrary, our method provides a fast and efficient solution for adjusting the editing level, which we consider a major contribution to downstream applications.
>
> Compositional generation [6,7,25] remains a challenging task for state-of-the-art generative models [2,30,35]. As shown in Section 5.3, our method significantly enhances the quality of compositional generation in a training-free manner.
>
> We would like to emphasize that the numerous papers on these two application areas accepted at NeurIPS highlight the broad applicability of such techniques. Additionally, there are also many papers working on interpolation in generative model as well, which can benefit various applications [6,7,11,25,38,49,50,59].
>
> > Q2: Based on proposition 1, text embedding interpolation can be viewed as manipulating the K/V which leads to suboptimal interpolation. However in (fused) inner-interpolation, still only K/V are manipulated.
>
> Indeed, both forms of interpolation manipulate the K/V.  However, the manipulation targets of these two methods are different. Text embedding interpolation (Sec. 3.4) only manipulates the K/V in the cross-attention layer. (Fused) inner-interpolation manipulates the K/V of both cross- and self-attention (Sec. 4.1). Furthermore, the fused version also concatenates the original and manipulated K/V to boost fidelity. This is mentioned in Sec.4.1 (L193-L197). We will clarify in the revision to emphasize these differences.
>
> > Q3: Is there a systematic way to determine the coefficient for Beta distribution, or is it a hyperparameter that needs tune for different models or prompts?
>
> The hyperparameter $\alpha$ and $\beta$ does need tuning for different prompts but can be selected automatically by Bayesian optimization as shown in the Appendix. D. In our experiments, the optimization starts from $\alpha=T/2$ and $\beta=T/2$ with 5 rounds where $T$ is the inference timesteps of diffusion model. In this case, the computation overhead is minimal for each sample.
>
>
> > Q4: How does applying interpolation module in self-attention or cross-attention alone affect the performance?
>
> Interpolating in cross-attention alone is equivalent to interpolating the text embedding (Sec. 3.4) and provides sub-optimal results (see Fig.2 and Tab.1 (a)).  Interpolating the self-attention alone leads to bad results similar to alpha interpolation. We will add one more section in the Appendix to discuss it in the revision.
>
> Additional references:
>
> [57] Sadeep Jayasumana, Srikumar Ramalingam, Andreas Veit, Daniel Glasner, Ayan Chakrabarti,and Sanjiv Kumar. Rethinking fid: Towards a better evaluation metric for image generation,2024.
>
> [58] Min Jin Chong and David Forsyth. Effectively unbiased fid and inception score and where to find them, 2020.
>
> [59] Mingdeng Cao, Xintao Wang, Zhongang Qi, Ying Shan, Xiaohu Qie, and Yinqiang Zheng. Masactrl: Tuning-free mutual self-attention control for consistent image synthesis and editing, 2023.
>
> Other references are notated as the same as main paper.

---

### Official Review · Reviewer_oKyj · 2024-07-14

**Soundness:** 3
**Presentation:** 3
**Contribution:** 3
**Rating:** 6
**Confidence:** 3

**Summary:**

In this paper the authors explore the task of interpolation between images in conditional diffusion modal. First the authors list out the three desirable properties of successful interpolation: perceptual consistency, smoothness, and image quality. The authors first introduce a method AID that incorporates three ideas: (1) interpolation should be done with both cross attention and self attention, (2) instead of a simple interpolation a fused interpolation should be performed, (3) the interpolation coefficients are sampled with a beta distribution instead of sampling uniformly.

The authors also introduce a second variation of their method called PAID (Prompt guided conditional interpolation) where the users can optionally add a prompt representing the intermediate image.

**Strengths:**

- The paper is well written and well organized. The authors do a good job first analyzing the issues with interpolating just the text prompt and then proposing a method that addresses them.

- The visual results shown in the paper look impressive and the authors show that the method can also be applied to other tasks like image editing and compositional generation.

- The authors use a comprehensive set of metrics that capture the 3 different aspects of good interpolation and show that the proposed method is helpful for each of the three metrics.

**Weaknesses:**

- One aspect of diffusion models that the authors have not considered here is the classifier free guidance and the use of negative prompts, which is standard practice for generating high quality images. Appendix H in supplement mentions that there are some results shown with negative prompt. But it would be useful to have a more detailed discussion of this in the main paper.
- The authors show an ablation study in Table 1(b). However the discussion for these experiments is very brief. It would be useful if the authors could show some visual results corresponding to these ablation experiments and a more thorough discussion.

(Minor):
- “text-to-diffusion” → “text-to-image” L94

**Questions:**

Please see the weaknesses above.

**Limitations:**

The authors have included a limitations section in the paper.

---

> ### Author Rebuttal · Authors · 2024-08-06
>
> We thank reviewer oKyj for the candid feedback and helpful suggestions. We provide our response to the reviewer's concerns in the following.
>
> > W1: One aspect of diffusion models that the authors have not considered here is the classifier free guidance and the use of negative prompts, which is standard practice for generating high quality images. Appendix H in supplement mentions that there are some results shown with negative prompt. But it would be useful to have a more detailed discussion of this in the main paper.
>
>
> Thanks for the suggestion to add a discussion on classifier-free guidance and the use of negative prompts. In our quantitative experiments, we use a fixed scale 7.5 of classifier-free guidance for all generated images without negative prompt. In appendix H, we use common negative prompt "monochrome, lowres, bad anatomy, worst quality, low quality" to boost visual quality. Generally, the effect of classifier-free guidance scale and negative prompt for our method is similar to the base generative model. We will add the discussion to main paper.
>
> > W2: The authors show an ablation study in Table 1(b). However the discussion for these experiments is very brief. It would be useful if the authors could show some visual results corresponding to these ablation experiments and a more thorough discussion.
>
> Thanks for this advice. We show the visual results of ablation study in Fig. 4 (a). As it shows, in the first row, pure attention interpolation will achieve high consistency but lack of fidelity. In the second row, fusing with self-attention can increase the fidelity. In the third row, selecting with Beta distribution can increase the smoothness further.

---

### Decision · Program_Chairs · 2024-09-25

**Decision:**

Accept (poster)

**Comment:**

This paper received somewhat mixed reviews.  The authors did not submit a rebuttal PDF but did provide rebuttal comments and participated in discussion with the reviewers.  The main concerns raised were around motivation and novelty.    The AC reviewed the paper in detail along with the reviews and discussion.  The AC disagrees with some of the reviewer and finds the proposed problem formulation to be interesting and potentially relevant given current interest in controllable image generation.  Further, as recognized by oKyj, the proposed method produces compelling results and is reasonably well evaluated.  As a result, the AC recommends acceptance.